# Aperiodic and oscillatory systems underpinning human domain-general cognition
Runhao Lu [1] ✉, Nadene Dermody[1], John Duncan [1] & Alexandra Woolgar [1,2]

Domain-general cognitive systems are essential for adaptive human behaviour, supporting various cognitive tasks through flexible neural mechanisms. While fMRI studies link frontoparietal network activation to increasing demands across various tasks, the electrophysiological mechanisms underlying this domain-general response to demand remain unclear. Here, we used MEG/EEG, and separated the aperiodic and oscillatory components of the signals to examine their roles in domain-general cognition across three cognitive tasks using multivariate analysis. We found that both aperiodic (broadband power, slope, and intercept) and oscillatory (theta, alpha, and beta power) components coded task demand and content across all subtasks. Aperiodic broadband power in particular strongly coded task demand, in a manner that generalised across all subtasks. Source estimation suggested that increasing cognitive demand decreased aperiodic broadband power across the brain, with the strongest modulations overlapping with the frontoparietal network. In contrast, oscillatory activity showed more localised patterns of modulation, primarily in frontal or occipital regions. These results provide insights into the electrophysiological underpinnings of human domain-general cognition, highlighting the critical role of aperiodic broadband power.

Human cognition is accomplished by the joint contributions of both domain-general and highly specialized neural systems[1–3]. A specific set of frontoparietal regions, collectively referred to as the multiple-demand (MD) network, is known to support domain-general cognition by being consistently co-activated during a variety of cognitive tasks, such as working memory, task switching, inhibitory control, and many more[4–9]. While the MD network's engagement across diverse tasks is well-established in fMRI studies, it remains unclear whether there are corresponding domain-general electrophysiological responses—measured using magnetoencephalography (MEG) or electroencephalography (EEG)—that show similar generalization across these tasks. Investigating this question is important, as electrophysiological signals directly reflect neuronal activity, providing an essential bridge between electrophysiological findings and fMRI results.

Neural activity obtained from MEG/EEG is a mixed signal consisting of both aperiodic (also referred to as "1/f-like" or fractal) and oscillatory components[10–12]. Although early studies treated aperiodic components as noise, current theories suggest that aperiodic activity may reflect widespread brain network excitability and the balance of excitatory and inhibitory neural activity (E/I balance)[13,14], and potentially underlie the network- and global-level signals observed in fMRI[15,16]. Recent computational work

highlights the importance of E/I balance as a fundamental principle of cortical circuits across a range of cognitive tasks, such as working memory and decision making[17–19]. Aperiodic activity, which serves as a non-invasive metric of E/I balance[13], has been found to play functional roles in various cognitive processes like perception, attention, working memory, memory consolidation, cognitive processing speed, and cognitive demand[10,11,14,20–25]. Notably, a recent study found that aperiodic activity outperformed oscillatory power and cross-frequency phase-amplitude coupling in indexing cognitive load variation during a counting task[25]. These findings highlight the significant relationship between aperiodic activity and fundamental cognitive processes, suggesting that E/I balance mechanisms may underpin these cognitive functions. However, it remains largely unknown whether aperiodic activity responds to task demand across multiple tasks (thus supporting domain-general cognition) and where demand-related aperiodic signals are located in the brain.

For oscillatory activity, many MEG/EEG studies have found that neural oscillations at different frequency bands are associated with task demand in various tasks[26–29]. Interestingly, a body of findings suggests that oscillatory power in many different frequency bands, such as theta (3–7 Hz), alpha (8–12 Hz), and beta (15–30 Hz), is modulated by task demand, with

[1]MRC Cognition and Brain Sciences Unit, University of Cambridge, Cambridge, UK. [2]Department of Psychology, University of Cambridge, Cambridge, UK. ✉e-mail: Runhao.Lu@mrc-cbu.cam.ac.uk

either increased or decreased power (e.g., theta[30–35]; alpha[36–41]; beta[42–45]). However, it is not yet clear to what extent these demand-related responses are similar between tasks, reflecting a domain-general property for cognitive control. Additionally, as most of the previous studies have not separated aperiodic and oscillatory activity, it is uncertain to what extent the reported results specifically reflect modulations of oscillatory activity in bands, or changes in global aperiodic activity, or both[10,12].

In addition to responding more strongly to difficult compared to easy cognitive tasks, the domain-general MD network is also known to encode a broad range of task-relevant information such as stimuli, rules, and responses[1,46–49]. This suggests that the MD network represents multiple aspects of a task, rather than simply reflecting cognitive effort[46]. Building on this, it is important to ask whether the electrophysiological counterparts of the domain-general system also exhibit this versatility of encoding. Specifically, beyond coding task demand, do these signals also encode other information, such as the task content, in a manner consistent with domain-general functionality?

To address these questions, we recorded neural signals using combined MEG/EEG while participants performed three cognitive tasks, each with two different sets of stimuli and two levels of demand. Using irregular resampling auto-spectral analysis[50] (IRASA), an analytical technique that separates aperiodic and oscillatory activity by averaging spectra generated from slightly shifted resampling factors, we extracted aperiodic components indexed by three parameters (broadband power, slope, and intercept) and oscillatory power in three canonical frequency bands (theta, alpha, and beta). We then asked whether any aperiodic or oscillatory components coded task demand (hard vs. easy) across all the subtasks using multivariate pattern analysis (MVPA), which identifies patterns in the neural data that differentiate experimental conditions (e.g., hard vs. easy)[51,52]. To explore the spatial distribution of these signals, we estimated their demand-related cortical source patterns and compared these patterns across different subtasks and oscillatory bands. Next, we investigated whether these responses in one subtask could be generalised to other subtasks, indicating a domain-general property. Finally, we examined whether these demand-related electrophysiological signals simultaneously code task content information (alphanumeric vs. colour) with similar source patterns, demonstrating the adaptive coding property akin to that of the domain-general MD network[1,46,48,49].

To preview, we found that both aperiodic and oscillatory components encoded task demand and content across all subtasks. Aperiodic broadband power, in particular, strongly coded task demand in in a manner that generalised across subtasks, with its source patterns partially overlapping the frontoparietal network. In contrast, oscillatory activity exhibited less generalisability and showed more localised source patterns, primarily in frontal or occipital regions.

## Results
### Behavioural results
Our primary goal was to investigate potential aperiodic and oscillatory signals reflecting change in task demand across multiple cognitive tasks. We therefore employed three cognitive tasks (Fig. 1A), which were a working memory task (WM), a switching task (SWIT) and a multi-source interference task (MSIT). Each task had two different stimulus contents (alphanumeric or colour stimuli) and two levels of demand (hard vs. easy). This design resulted in six subtasks (3 tasks * 2 contents).

To verify the demand manipulation, we conducted 2 (task demand: hard vs. easy) * 2 (task content: alphanumeric vs. colour) repeated measures ANOVAs on behavioural accuracy and reaction time (RT) for each task. Indeed, there was a significant main effect of task demand on accuracy for all three tasks, showing that participants were more accurate in easy conditions than in hard conditions [Fig. 1B WM: $F(1,42) = 106.63$, $p < 0.001$, $\eta_p^2 = 0.72$; SWIT: $F(1,42) = 11.49$, $p = 0.002$, $\eta_p^2 = 0.22$; MSIT: $F(1,42) = 54.66$, $p < 0.001$, $\eta_p^2 = 0.57$; see Table S1 for descriptive statistics]. Similarly, there was a significant main effect of task demand on RT for all three tasks, showing that participants had slower RT in hard conditions

compared to easy conditions [WM: $F(1,42) = 207.15$, $p < 0.001$, $\eta_p^2 = 0.83$; SWIT: $F(1,42) = 114.11$, $p < 0.001$, $\eta_p^2 = 0.73$; MSIT: $F(1,42) = 437.87$, $p < 0.001$, $\eta_p^2 = 0.91$; see Table S1 for descriptive statistics]. These results confirmed the validity of task demand manipulation across all tasks.

We then examined whether participant's accuracy and RT varied with task content. For behavioural accuracy we did not find a significant content effect or interaction between task demand and content on accuracy in the SWIT [content effect: $F(1,42) = 0.01$, $p = 0.94$; interaction: $F(1,42) = 0.02$, $p = 0.90$] or MSIT tasks [content effect: $F(1,42) = 1.11$, $p = 0.30$; interaction: $F(1,42) = 3.67$, $p = 0.06$]. However in the WM task, we found a significant main effect of task content [$F(1,42) = 21.80$, $p < 0.001$, $\eta_p^2 = 0.34$] and a significant interaction between task demand and task content [$F(1,42) = 31.69$, $p < 0.001$, $\eta_p^2 = 0.43$]. Participants were significantly less accurate in the colour WM task than that in the alphanumeric WM task in the hard condition [$F(1,42) = 32.49$, $p < 0.001$] but not in the easy condition [$F(1,42) = 0.20$, $p = 0.65$]. For RT, we found a significant content effect in the WM tasks [$F(1,42) = 50.16$, $p < 0.001$, $\eta_p^2 = 0.54$] and the MSIT tasks [$F(1,42) = 5.83$, $p = 0.02$, $\eta_p^2 = 0.12$]. Participants had slower RT in the colour compared to alphanumeric MSIT tasks, and they had faster RT in the colour compared to alphanumeric WM task. The WM result may have reflected a speed-accuracy trade-off in the WM tasks, in that participants tended to respond faster in the colour task but with lower accuracy (at least in the hard condition). We did not find a significant content effect on RT in the SWIT tasks [$F(1,42) = 0.46$, $p = 0.50$] or interactions between demand and content in any tasks [all $F$s < 1.84, $p$s > 0.18].

The behavioural results therefore indicated that hard conditions were more demanding than easy ones across all tasks, as intended. Task performance was similar for the alphanumeric and colour versions of each task, with the exception of the WM task in which there appeared to be a speed-accuracy trade-off in which participants were slightly faster and substantially less accurate in the colour version of the high demand task. Participants also tended to respond more quickly in the alphanumeric MSIT task compared to the colour MSIT task.

### Domain-general coding of task demand by aperiodic components
To obtain the aperiodic and oscillatory neural components in MEG/EEG data, we first subtracted evoked potentials of each condition from the MEG/EEG signal to remove evoked and phase-locked activity[32]. Then, we applied the IRASA[50] on the time windows of 0.3–1.5 s from stimulus onset for each task to separate the aperiodic and oscillatory components from the mixed signal (Fig. 2A). We excluded the initial 0–0.3 s to focus on the neural dynamics more directly related to cognitive demand, as the early window may primarily reflect perceptual activity. For the WM tasks, excluding this period allowed us to specifically analyse the maintenance period rather than the stimulus presentation period.

First, we used MVPA to ask whether we could decode task demand based on three parameters that describe the aperiodic component of the signal (3–30 Hz broadband power, slope, and intercept) across MEG/EEG sensors. Indeed, as shown in Fig. 2B, we could decode task demand from all three aperiodic parameters across all subtasks with significant above-chance areas under the receiver operating characteristic curve (AUC) (all $t$s > 5.19; FDR-corrected $p$s < 0.001). For all signals, the AUC was relatively high in the WM and MSIT tasks but lower in the SWIT tasks. Aperiodic broadband power, which reflects both the slope and intercept, tended to show the highest AUC.

We then estimated the cortical sources of these demand-coding aperiodic signals. For this, we estimated the source space responses for each individual, parcellated into 360 regions of interest (ROIs; 180 regions per hemisphere without the medial wall) based on the Human Connectome Project multimodal parcellation (HCP-MMP1.0)[53]. We then re-ran the demand decoding analysis using the activity in all these source-level ROIs and transformed the resulting classifier weights, by multiplying them with the covariance of the data, to yield interpretable values reflecting the signal contributed by each source ROI to the classification[54] (see Methods).

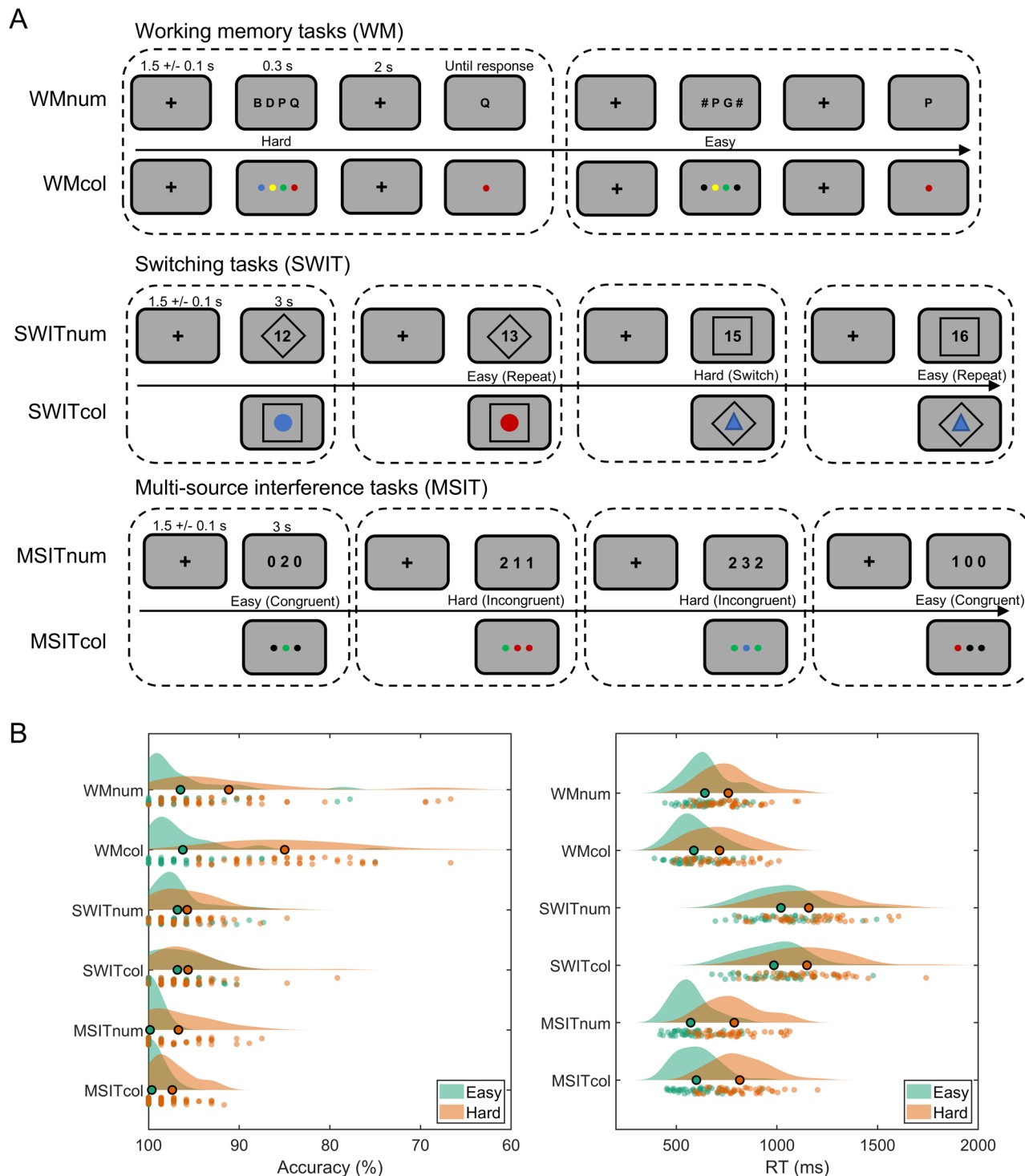

**Fig. 1 | Experimental paradigm and behavioural results. A** Experimental design of the working memory task (WM), the switching task (SWIT), and the multi-source interference task (MSIT). Each task had two versions with either alphanumeric or colour stimuli as task contents. For the WM task, participants were required to remember either four items (hard condition) or two items (easy condition). The items were letters in the alphanumeric condition and were coloured circles in the colour condition. For the SWIT task, participants made responses based on the current rule indicated by the shape (a square or a diamond) that surrounded the item. Switch trials (the rule for the present trial differed from the last trial) were considered the hard condition, and repeat trials (the rule for the present trial repeated the last trial) were considered the easy condition. For the MSIT task,

participants needed to identify the unique item among three presented items. In the easy (congruent) condition, the target item was presented in the position compatible with its original value (e.g., "100" or "red, black, black"). In the incongruent (hard) condition, the target item was presented in the position incongruent with the original value and always flanked by different interfering numbers or colours (e.g., "331" or "blue, blue, red"). **B** Behavioural accuracy (left) and response time (RT; right) in easy and hard conditions for each subtask. Each individual dot in the raincloud plots represents a participant and the bolded dot shows the mean. Repeated measures ANOVAs (task demand * task contents) showed that the main effects of task demand (hard vs. easy) were significant for both accuracy and RT for all 6 subtasks (all $ps < 0.002$).

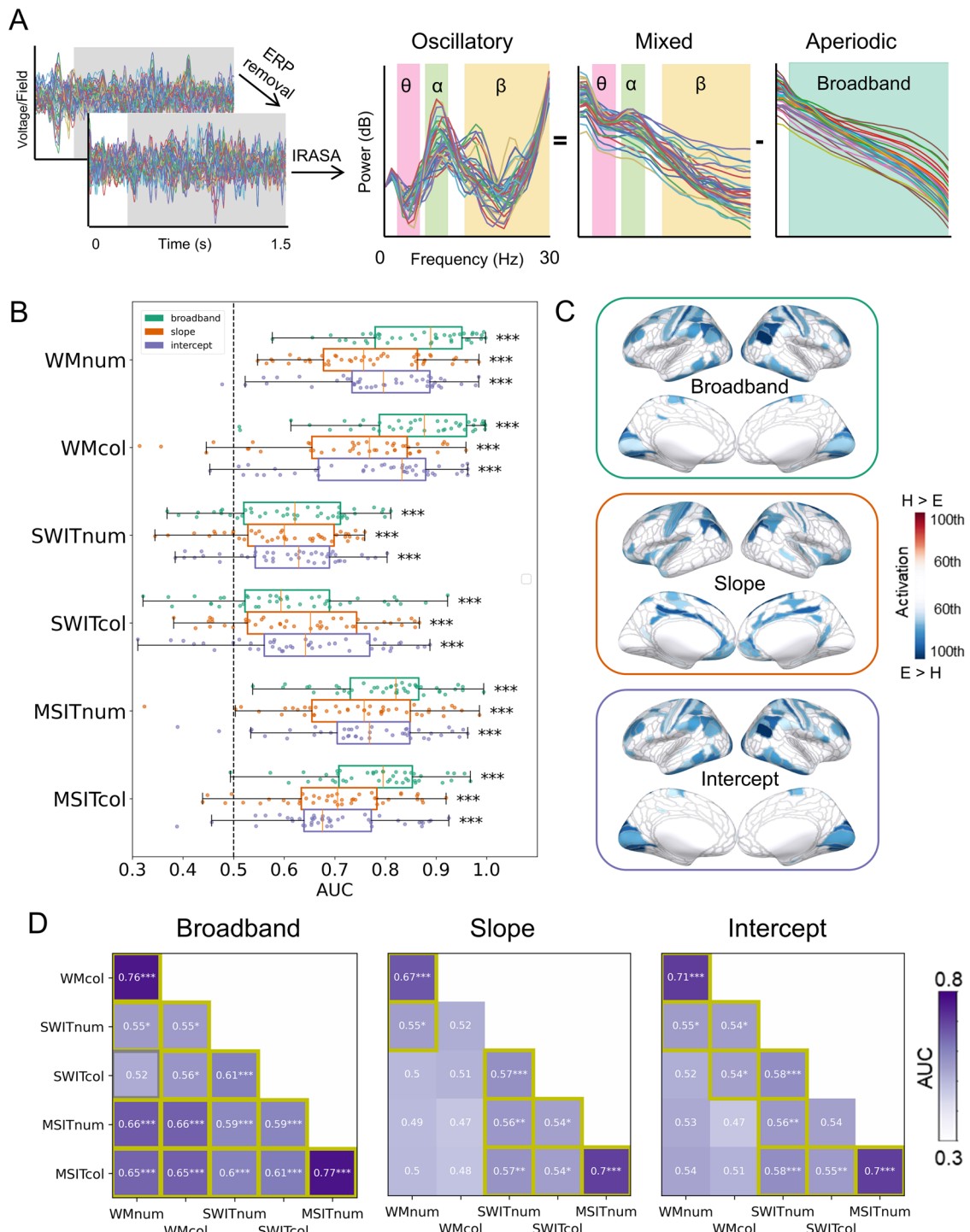

Positive values indicated increased activity in the hard condition compared with the easy condition. To illustrate the sources contributing to the decoding of task demand, we averaged the patterns across all the six subtasks for each signal and visualised the most informative regions, corresponding to the 60th to 100th percentiles, on an inflated template brain (Fig. 2C, see also Fig. S1A for the full unthresholded map).

The most informative regions underpinning demand coding in aperiodic broadband power and intercept were similar to one another (Fig. 2C, top and bottom panel). They showed a widely distributed spatial pattern with decreases in lateral frontoparietal MD-like regions, motor regions, and visual regions. The most informative regions for aperiodic slope were somewhat different from the other two aperiodic parameters, mainly

located in the cingulate cortex, motor regions, and parietal regions. This spatial pattern of demand-related aperiodic activity was highly consistent across the six subtasks (Fig. S1B), suggesting a domain-general aperiodic responses to increased cognitive demand.

Next, we quantified the extent to which the effect of demand was comparable across subtasks. To address this, we performed a cross-task generalisation of demand classification in sensor space, where we by turns trained classifiers to distinguish task demand on one subtask and tested them on other subtasks (see Methods). If task demand has a similar effect on activity across subtasks, we should see above-chance AUC for cross-task generalisations. As shown in Fig. 2D, we found consistent above-chance generalisability for broadband power across all different subtasks except

**Fig. 2 | Domain-general coding of task demand by aperiodic components. A** To separate the aperiodic and the oscillatory components from the mixed signal, we first subtracted the event-related potentials from the timeseries data for each condition to remove the phase-locked evoked signals. Then, we used irregular resampling auto-spectral analysis (IRASA) based on the time window of 0.3–1.5 s from stimulus onset to obtain the aperiodic components in the frequency domain. After subtracting the aperiodic components from the mixed activity, we obtained the oscillatory components in the frequency domain. We selected theta (3–7 Hz), alpha (8–12 Hz), and beta (15–30 Hz) as the frequencies of interest for further analyses. For the aperiodic components, we used the broadband power (3–30 Hz), slope, and intercept for further analyses (both slope and intercept were obtained from the linear function that best fitted the aperiodic power spectrum) (**B**) Decoding results on task demand (hard vs. easy) using aperiodic activity for each subtask. All MEG/EEG sensors were used for decoding. Each dot means one participant. Error bars represent standard errors. **C** Source estimation patterns for demand decoding averaged across all the subtasks for aperiodic signals. Outlines show 360 cortical regions based on the Human Connectome Project multimodal parcellation (HCP-MMP1.0)[53]. Coloured regions represent the 60th to 100th percentiles of activation (hard vs. easy discrimination) across the brain (H: hard; E: easy). Negative (blue) values indicate decreased activity (or a steeper slope) in the hard condition compared to the easy condition. The full map and source estimation patterns for each subtask separately are shown in Fig. S1. **D** Cross-task generalisation of task demand coding based on aperiodic activity. All MEG/EEG sensors were used for decoding. Classifiers were trained in one subtask and then tested in other subtasks with the same signal. Each coloured box represents the average of the generalisation performance between two paired train-test schemes within a pair of subtasks (e.g., training on A, testing on B and training on B, testing on A), with significantly above-chance AUC (highlighted with yellow borders) meaning the signal was generalisable across the two tasks. WM: Working memory task; SWIT: Switching task; MSIT: Multi-source interference task; num: Alphanumeric task; col: Colour task. $^*p < 0.05$, $^{**}p < 0.01$, $^{***}p < 0.001$ (FDR-corrected).

---

between the colour SWIT and alphanumeric WM tasks. Thus, with one exception, the effect of demand on broadband aperiodic power was demonstrably similar across the three different cognitive tasks and their stimulus variations. The slope and intercept parameters of the aperiodic components showed generalisability between different contents of the same task (e.g., WM tasks with alphanumeric or colour contents), and between SWIT and MSIT across tasks and contents.

In summary, we found that aperiodic activity coded task demand across all subtasks with decreased aperiodic responses to increasing cognitive demand across widespread sources irrespective of the particular cognitive task or stimulus content. Moreover, a classifier showed strong cross-task generalisation of aperiodic activity, especially as captured by broadband power, in the pattern of change over subtasks. These results suggest that aperiodic activity could play a domain-general role in supporting cognition.

### Coding of task demand by oscillatory power
We then asked how demand modulated oscillatory responses. For this we used MVPA to quantify whether oscillatory power in different frequency bands (theta: 3–7 Hz; alpha: 8–12 Hz; beta: 15–30 Hz) was modulated by task demand. As above, we used data calculated from all the MEG/EEG sensors and performed the demand decoding analysis on each subtask separately. As shown in Fig. 3A, we found that oscillatory power in all three frequency bands could code task demand across all subtasks with significantly above-chance AUC (all $ts > 3.22$; FDR-corrected $ps < 0.002$). AUC was again relatively high in the WM and MSIT tasks compared to the SWIT task. Interestingly, oscillatory components tended to show lower AUC than aperiodic components in general.

We then estimated the source patterns of demand-related oscillatory power in each frequency band (Fig. 3B and Fig. S2A). We found increased demand-related theta power in the medial frontal regions under hard conditions, along with a decrease in theta power in occipital regions. Alpha power, instead, showed an increase in the occipital regions. Beta power mainly showed an increase in the lateral and medial frontal regions.

To quantitatively examine the inter-correlations among the three oscillatory components, we correlated the subtask-averaged source patterns over the 360 parcels in each individual separately, and then compared the distribution of resulting Pearson's r values to chance using permutation tests (Fig. 3C). The distribution of activation patterns over spatial sources was significantly anticorrelated between theta and alpha (mean $r = -0.77$ across participants, $p < 0.001$ with 1000 permutations), correlated between theta and beta (mean $r = 0.38$ across participants, $p < 0.001$ with 1000 permutations), and anticorrelated between alpha and beta (mean $r = -0.48$ across participants, $p < 0.001$ with 1000 permutations).

When separately examining the source patterns for each subtask (Fig. S2B), we found that demand-related oscillatory power showed broadly consistent spatial patterns across subtasks, with minor specific patterns differentiating the subtasks. For example, theta power had a stronger demand-related increase in medial frontal regions during the SWIT and

MSIT but not WM tasks, while alpha power showed a demand-related increase in motor regions that was more pronounced in the two WM subtasks than the other tasks.

In addition, cross-task generalisation results indicated that the three bands of oscillatory power exhibited generalisability across some but not all subtasks (Fig. 3D). In particular, they all showed high generalisability between different contents of the same task. Additionally, oscillatory power in all three frequency bands generally showed significant generalisability between MSIT and SWIT tasks, with a couple of exceptions, but tended to show a lack of generalisation between WM and SWIT, and for beta between the alphanumeric WM task and any version of SWIT or MSIT. These findings suggested that the oscillatory components might code task demand through a combination of shared and distinct patterns across tasks. These codes are generalisable for some tasks but may change in other specific tasks.

To summarize, these results indicated that oscillatory power in theta, alpha, and beta bands were also modulated by task demand with fairly consistent source patterns across subtasks but with some task-based idiosyncrasies. Although oscillatory power showed good cross-generalisability for some tasks, decoding tended to be weaker than we had seen for aperiodic activity (especially broadband aperiodic power) and could not generalise across all the tasks, suggesting that there might be both shared and task-specific responses to different types of cognitive demand.

### Coding of task content by aperiodic and oscillatory components
In addition to responding to a range of different task demands, a domain-general system may be expected to code task-relevant information across multiple tasks[47,55]. Therefore as a final set of analyses, we examined whether aperiodic and oscillatory components of the MEG/EEG signals also coded task content information (alphanumeric vs. colour).

As shown in Fig. 4A, B, we found that, in sensor space, both aperiodic and oscillatory components showed significant above-chance AUC for decoding task content across the three tasks (aperiodic: $ts > 15.31$; FDR-corrected $ps < 0.001$; oscillatory: $ts > 4.77$; FDR-corrected $ps < 0.001$), indicating that all these signals also reflected task content. The decoding profiles were similar across different tasks for both aperiodic and oscillatory results. For aperiodic components, broadband power again showed the highest AUC, while intercept showed the lowest but still above-chance AUC. For oscillatory power, numerically beta band showed the highest AUC, while theta band showed the lowest but still above-chance AUC. In general, the aperiodic components again tended to show higher AUC than the oscillatory components.

Next we examined the source distribution of the patterns. As shown in Fig. 4C, D, we found that the cortical patterns that contributed to task content classification were visually similar to those found in decoding task demand (Figs. 2C and 3B). To quantify this, we calculated Pearson's correlations across the 360 ROIs between task-averaged demand and content decoding patterns for both aperiodic and oscillatory signals for each individual, and then compared the resulting distribution of r values to chance

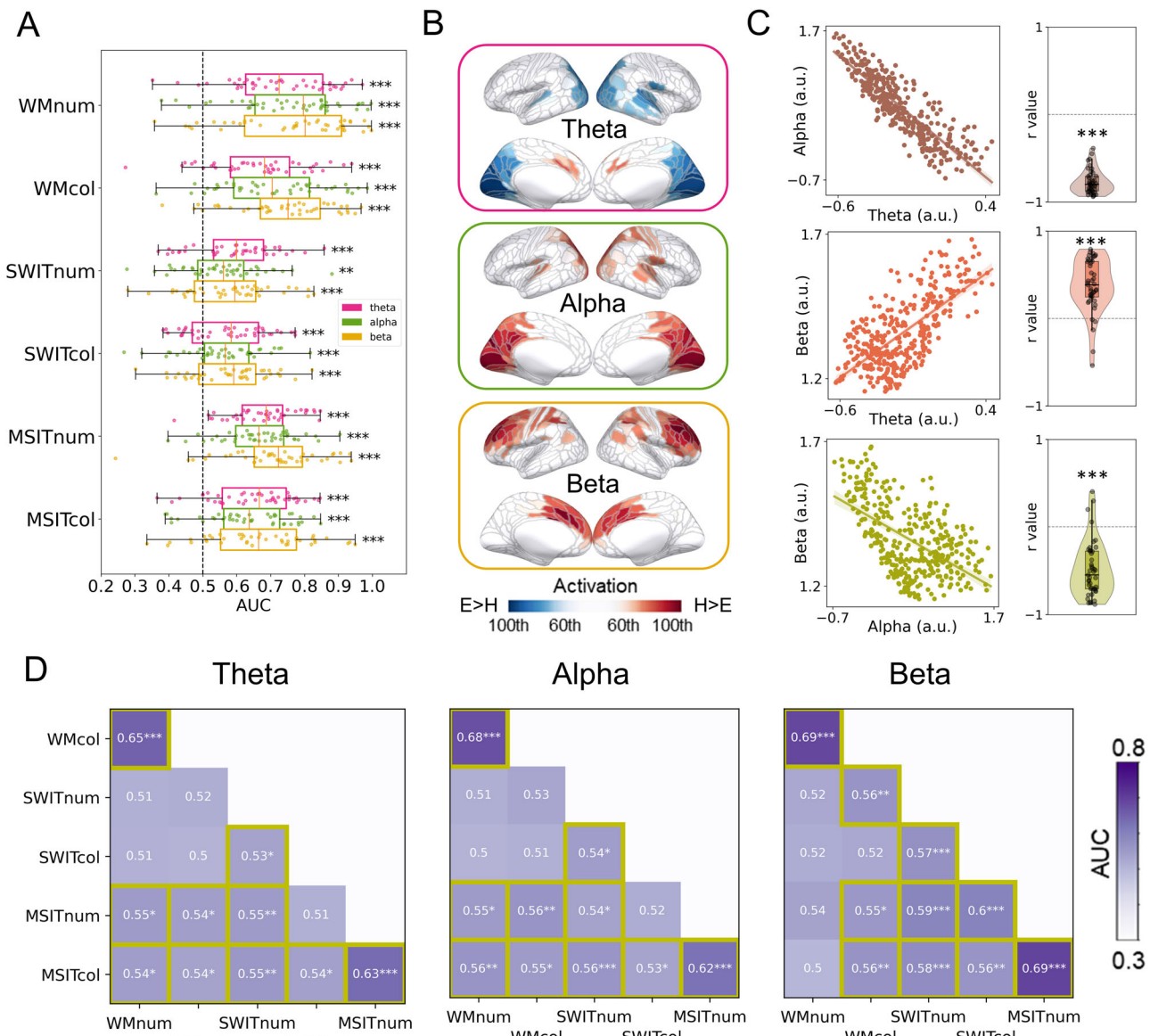

**Fig. 3 | Coding of task demand by oscillatory power. A** Decoding of task demand (hard vs. easy) using oscillatory activity for each subtask. All MEG/EEG sensors were used for decoding. Each dot means one participant. Error bars represent standard errors. *** $p < 0.001$ (FDR-corrected). **B** Source estimation patterns for demand decoding averaged across the subtasks for the oscillatory signals. Coloured regions represent the 60th to 100th percentiles of activation (hard vs. easy discrimination) across the brain (H: hard; E: easy). Positive (red) values indicate increased activity in the hard condition compared to the easy condition. Source estimation patterns for the full map (0th to 100th percentiles) and for each subtask separately are shown in Figure S2. **C** Pearson's correlation between source estimation patterns (360 ROIs in source space) that coded task demand from oscillatory power in different frequency bands. Left: Scatter plots using data averaged across subtasks and participants for illustration. Each dot represents a cortical ROI. Right: Correlation coefficients from within-subject analyses, each dot represents a single participant. *** $p < 0.001$ with 1000 permutations (**D**) Cross-task generalisation of task demand using oscillatory activity from all MEG/EEG sensors. Classifiers were trained on one subtask and then tested on other subtasks with the same signal. The results show the generalisation between different tasks, with significantly above-chance AUC (highlighted with yellow borders) meaning the signal was generalisable across the two tasks. Abbreviations as in Fig. 2. * $p < 0.05$, ** $p < 0.01$, *** $p < 0.001$ (FDR-corrected).

using 1000 permutations (Figure S3; see Methods). Results showed that demand and content decoding patterns in source space were significantly correlated for both aperiodic (mean $r = 0.82$, 0.66, and 0.83 across participants, respectively, for broadband power, slope, and intercept; all $ps < 0.001$ with 1000 permutations) and oscillatory signals (mean $r = 0.81$, 0.81, and 0.72 across participants, respectively, for theta, alpha, and beta; all $ps < 0.001$ with 1000 permutations). Specifically, the source patterns of aperiodic broadband power and intercept for classifying task content again showed a wide range of cortical contribution across the frontoparietal MD-like regions and the cingulo-opcular regions. The aperiodic slope mainly showed spatial patterns in frontal, cingulate, and parietal regions. For oscillatory power, theta power that responded to task content mainly originated from

visual regions; alpha power showed a reversed patterns to theta but also located in the occipital regions; and beta power mainly came from the lateral and medial prefrontal regions. We also separately examined the source patterns for each task (Figure S4), and they showed similar patterns to the averaged pattern across tasks.

To further understand the inter-correlations among oscillatory power based on their source patterns across 360 cortical regions, we again performed Pearson's correlations (Fig. 4E). As for demand, the distribution of activation patterns over spatial sources that supported content coding was significantly anticorrelated between theta and alpha, correlated between theta and beta, and anticorrelated between alpha and beta (mean $r = -0.65$, $-0.28$, and $-0.54$ across participants, respectively; all $ps < 0.001$ with 1000 permutations).

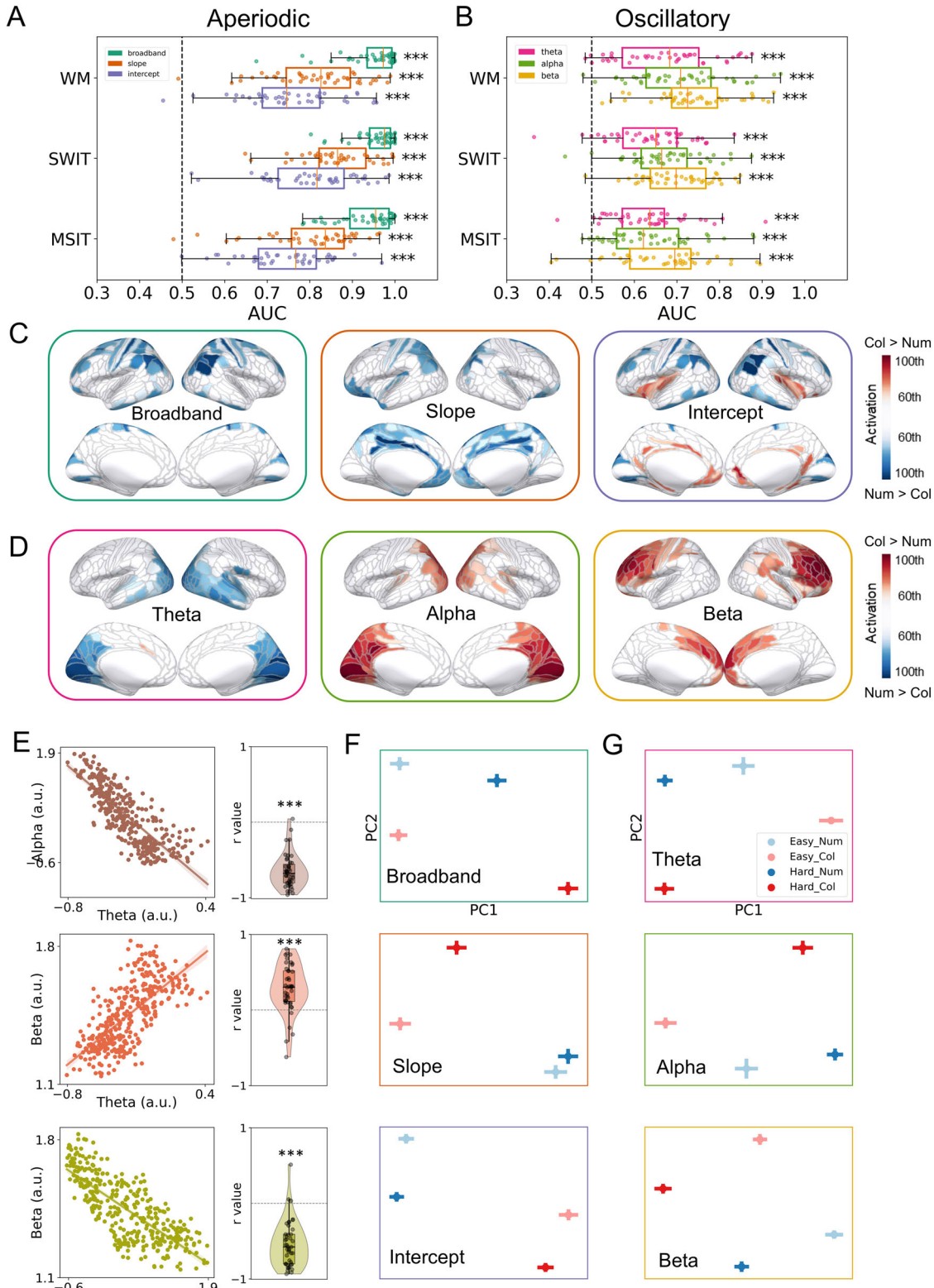

**Fig. 4 | Aperiodic and oscillatory components code task content. A, B** Decoding of task content (alphanumeric vs. colour) using (**A**) the aperiodic and (**B**) the oscillatory activity for each task. All MEG/EEG sensors were used for decoding. Each dot means one participant. Error bars represent standard errors. ***$p < 0.001$ (FDR-corrected). **C, D** Source estimation patterns for content decoding averaged across all tasks for (**C**) aperiodic and (**D**) oscillatory signals. Coloured regions represent the 60th to 100th percentiles of activation (colour vs. alphanumeric discrimination) across the brain. Positive values (red) indicate increased activity (or a shallower slope) in the colour condition compared to the alphanumeric condition. Source estimation patterns for each task separately are shown in Figure S4. **E** Pearson's correlation results between source estimation patterns (360 ROIs in source space) that coded task content from oscillatory power in different frequency bands. Left: Scatter plots using data averaged across subtasks and participants for illustration. Each dot represents a cortical ROI. Right: Correlation coefficients from within-subject analyses, each dot represents a single participant. ***$p < 0.001$ with 1000 permutations. **F, G** Representations of task demand and task content in two-dimensional space extracted from (**F**) the aperiodic and the (**G**) oscillatory signals from all MEG/EEG sensors. Abbreviations as shown in Fig. 2.

It should be noted that the behavioural accuracies of the alphanumeric tasks and the colour tasks were not strictly matched, especially for the hard condition in the WM tasks. To exclude the potential confounds of task demand in classifying contents, we repeated the analysis using only easy trials to decode task content, as the behavioural accuracies in the easy conditions were well matched across subtasks. The decoding results based on easy trials only, as well as their source estimations, showed very similar patterns to the results above (Figure S5).

Since both task demand and content were simultaneously coded by aperiodic and oscillatory components, we further visualised their representational geometry in two-dimensional space based on all MEG/EEG sensors. We averaged trials within each condition (12 conditions in total: 3 tasks * 2 demands * 2 contents) for each sensor and each signal, then performed principal component analysis (PCA) to extract the first two principal components (PCs) for visualization (see Methods). Figure 4F, G show the representations of task demand and task content (averaged across three tasks) in each signal. For both aperiodic and oscillatory signals, there was a clear boundary between the alphanumeric tasks (shown in blue) and the colour tasks (shown in red). Also, there was clear separation of easy (shown in light colours) and hard (shown in dark colours) trials for each subtask. These results indicate that although both task demand and content could be decoded from aperiodic and oscillatory signals, their coding dimensions appeared to be distinct.

Taken together, these results indicate that both aperiodic and oscillatory components reflect task demand and task content across multiple tasks. Source estimation results showed that the cortical patterns in support of task content classification were generally similar to those supporting task demand classification. Moreover, representational geometry results suggested that aperiodic and oscillatory systems code demand and content with different coding dimensions. These results together indicate that the same domain-general aperiodic and oscillatory systems can simultaneously code various task-relevant information, likely using different coding dimensions to support cognition.

## Discussion

Using MEG/EEG recording in combination with IRASA and MVPA, the present study asked whether the human brain exhibits domain-general aperiodic and/or oscillatory responses to multiple types of cognitive challenge. We recorded MEG/EEG signals when participants performed three different cognitive tasks (the WM, SWIT, and MSIT tasks) each with different levels of demand (hard vs. easy) and content (alphanumeric vs. colour). Decoding results showed that both aperiodic (broadband power, slope, and intercept) and oscillatory (in theta, alpha, and beta) components were modulated by task demand and content across all subtasks, with the patterns of demand modulation generalizable among the different tasks. Aperiodic broadband power showed the strongest cross-subtask generalisability, suggesting that it may reflect a domain-general mechanism supporting cognitive control. These results echo previous studies that showed modulation of aperiodic and oscillatory activity with task demand[25–28], and particularly emphasise the relevance of aperiodic components of electrophysiological activity.

Our findings provide support for the MD theory[1,5] by demonstrating that a widely distributed domain-general system can be detected in electrophysiological responses. This system, mainly reflected in aperiodic broadband power, exhibits robust generalization across tasks and coding of both task demand and task content. These results extend previous work in fMRI suggesting that human cognition is implemented by both domain-general and domain-specific systems[2,3], which collaboratively enable the flexible cognitive control required for diverse tasks. Additionally, our findings align with the adaptive coding hypothesis, which posits that domain-general systems flexibly encode various types of task-relevant information[46,47]. In particular, we observed that domain-general aperiodic activity and oscillatory power can represent both task demand and task content, albeit along different coding dimensions. This distinction highlights the adaptive capacity of domain-general systems to dynamically

represent multiple aspects of a task, supporting the hypothesis that these systems function as a flexible neural substrate for cognitive control.

The aperiodic 1/f-like spectral pattern has been observed ubiquitously in nature and across many different modalities of neural activity[11,23,56]. Although aperiodic activity was previously considered to reflect noise, recent computational work has suggested that it reflects the synaptic balance between excitation and inhibition, with more negative (steeper) aperiodic slope or power indicating a lower E/I ratio (i.e., increased inhibition)[10,13,57]. E/I balance is proposed to be a fundamental principle of cortical circuits supporting various of cognitive tasks, while disruptions in E/I balance may link to worse cognitive performance[18,19]. The mechanism of E/I balance that underlies aperiodic activity may explain its generalisable demand- and content-related representation across a variety of cognitive tasks. Empirically, an increasing number of studies have found roles for aperiodic activity in a wide range of cognitive processes including perception, attention, working memory, cognitive load, aging-related cognitive changes, and more[11,14,20–25,58–61]. One recent study compared the role of aperiodic activity and alpha oscillations in predicting processing speed, and found it was the aperiodic rather than oscillatory alpha activity that predicted processing speed[22]. Another study associated different EEG indices with cognitive load in a counting task, finding that aperiodic activity outperformed both oscillatory power and cross-frequency phase-amplitude coupling[25]. Our results are consistent with these findings and suggest that aperiodic activity coded both demand and content in a way that generalised across multiple tasks, suggesting a domain-general property. In addition, we observed that the aperiodic codes for demand and context were highly structured, with apparently orthogonal dimensions coding for content and demand (see Fig. 4F, G).

To examine the potential cortical distribution of demand- and content-related aperiodic activity, we performed source estimation and found a widely distributed pattern of aperiodic activity across many brain regions. Interestingly, the patterns of aperiodic broadband power and intercept partly overlapped with the frontoparietal MD regions, which may potentially link the domain-general MD network with domain-general electrophysiological signals. The frontoparietal distribution of aperiodic signals align with previous research[62], and might suggest that E/I balance in frontoparietal regions may particularly play an important role in supporting domain-general cognition[19]. Notably, although brain activity measured by fMRI and positron emission tomography (PET) in the MD network increase with task demand[2,6,63], we found that aperiodic broadband power in frontoparietal cortex decreased with task demand. This result is in line with previous findings in EEG that aperiodic intercept was decreased in high cognitive load conditions[10,25], potentially suggesting a relationship between fMRI activation and E/I balance. Consistent with this, a recent study using simultaneous resting-state EEG-fMRI found that aperiodic EEG components were negatively correlated with BOLD activation in frontal and parietal regions[16]. These findings highlight how MEG/EEG measures of aperiodic activity may complement fMRI or PET by offering a window into neuronal dynamics, and potentially shifts in E/I balance, that may underlie domain-general cognition. While both modalities capture domain-general systems, they may reflect distinct yet interconnected processes: fMRI and PET measure metabolic demands and regional activations, whereas MEG/EEG provides insights into electrophysiological activity including excitation and inhibition. Together, these perspectives enhance our understanding of how domain-general systems operate across multiple aspects of neural activity.

MEG/EEG oscillatory power has also been linked to various cognitive processes[26,44,64,65]. Although many studies have attempted to associate oscillatory power in different frequency bands with task demand, there remains ongoing debate about how oscillations in different bands are modulated by demands of various tasks, particularly regarding their cortical distributions and modulation directions[26–28]. One key factor contributing to this discrepancy is the potential confound of the aperiodic component[10,12,66]. In this study, we separated oscillatory activity from the aperiodic background and systematically examined the role of pure oscillatory power in

coding demand across multiple tasks. We found that with increasing task demand, theta power increased in the medial frontal regions and anterior cingulate cortex (ACC). This aligns with the classic theory about mid-frontal theta in cognitive control[26] and empirical studies showing that mid-frontal theta increases with higher task engagement[31,33,67–69]. Additionally, we observed a decrease in theta power in the occipital regions with higher demand, which potentially reflects information exchange between frontal and occipital regions[70,71].

Compared with theta, how alpha and beta power are modulated by task demand remains debated, with studies often showing contradictory evidence regarding the direction of change[27,28]. By examining the decoding from pure oscillatory power, we found alpha power increased with task demand in the occipital regions, consistent with previous findings on the load effect on occipital alpha increase[72,73]. A simultaneous EEG-fMRI study similarly showed that alpha power that increased with cognitive load originated in the early visual areas[74]. The increased alpha oscillations in visual areas might reflect the functional inhibition of early visual areas that are not currently required for the task[75]. For beta power, we found an increase in frontal regions with increasing task demand, consistent with a series of previous findings[43–45]. Intracranial studies have linked beta oscillations in local field potentials to top-down cognitive control supporting behaviour[42,76,77]. An increase in frontal beta oscillations may thus reflect enhanced recruitment of top-down control. Together, our findings provide insights into the debate on the relationship between oscillatory power and task demand by isolating pure oscillatory power from the aperiodic background. Moreover, by examining multiple tasks, we provide evidence that theta, alpha, and beta power can code both demand and content across many tasks. However, their coding strength and cross-task generalisability tended to be weaker compared to aperiodic activity.

Interestingly, we found that the source distributions for both aperiodic and oscillatory components in content coding were broadly similar to the patterns for demand coding. This finding highlights their domain-general properties, indicating that these components, arising in similar regions, can simultaneously code multiple aspects of task-relevant information. This is consistent with the observations in previous fMRI and non-human primate results, which showed that the domain-general MD network could flexibly code diverse information including visual, auditory, rule, and response[1,47–49]. However, when examining their representation geometry in coding both demand and content, we found clear and distinct organising axes for decoding demand and content, suggesting that the spatial patterns were subtly different. This result confirmed that the above-chance content decoding was not merely a consequence of different demands in the tasks, and further suggested that, despite arising from broadly similar regions, these signals may code different task-relevant information using distinct coding dimensions, perhaps to avoid inference[78,79]. The structured organisation of the aperiodic response to demand and content also lends credence to the idea that the changes in aperiodic power do not simply reflect a dampening down of random or unstructured noise (e.g., reduced noise when focusing) but suggests that aperiodic activity may reflect a meaningful neural signal (i.e., E/I balance) coding multiple task elements.

As oscillatory power in different frequency bands all contributed to demand and content decoding, we also examined the inter-relationships between the different bands. For both demand and content decoding, we found that the spatial distribution of theta power showed a positive correlation with beta power, while both theta and beta patterns showed negative correlations with alpha. In particular, with increased task demand, theta showed increase in medial frontal regions and decrease in occipital regions, while alpha showed a reversed pattern. Beta generally showed increase across the brain with increased demand, with the strongest increase in frontal regions. Previous studies have reported contradictory results regarding the relationship between theta and alpha, with many studies showing reversed patterns between them as we did[80,81], but some studies showed comparable patterns between them[72,82]. A recent study found that the cortical patterns of theta and alpha power in coding perception-action representations were generalisable across these two frequency bands,

indicating their potential functional similarities[82]. However, as mentioned above, most of these studies did not separate oscillatory and aperiodic components before examining the relationship between different frequency bands, which makes the results challenging to interpret. Due to the homogeneity of the broadband aperiodic power, oscillations in different frequency bands may tend towards showing positive correlations if the aperiodic components are not separated out. After separating aperiodic components from the oscillatory activity, our results provided more compelling evidence that the spatial distributions of theta and beta power are positively correlated, while both are negatively correlated with alpha power.

We acknowledge several limitations in this study. First, although the sample size was sufficient for group-level statistical analyses, we had limited ability to address questions regarding individual differences. Recruiting a larger and more diverse group of participants in future studies could provide a more nuanced understanding of variability across individuals. For instance, factors such as fluid intelligence may influence attentional allocation policies and thereby modulate the observed domain-general responses[69,83,84]. Exploring such relationships in larger samples could help uncover how individual differences shape these neural patterns. Second, while the cognitive tasks we used captured a range of cognitive control processes, they do not capture the full range of cognitive domains. Incorporating more diverse and ecologically valid tasks could strengthen the applicability of these findings to broader cognitive contexts. Third, the limited spatial resolution of MEG/EEG constrained the precision of our source localization. This underscores the importance of integrating multi-modal methods, such as fMRI, to enhance spatial specificity. Finally, although there is evidence linking aperiodic activity to E/I balance, finer mechanistic interpretations on these findings remain challenging. Addressing this will require dedicated computational modelling and systematic experiments at both intracranial and extracranial recording levels.

Building on our findings, several future directions could further validate and extend this work. A critical next step involves using multimodal approaches to deepen our understanding of the mechanisms underlying aperiodic and oscillatory activity. For instance, comparing MEG/EEG results with intracranial EEG data could provide crucial insights. The hypothesis linking aperiodic activity to E/I balance was largely developed with local field potential data, and confirming comparable patterns between intracranial EEG and MEG/EEG would strengthen the validity of non-invasive measures in assessing E/I balance and its role in cognition. Additionally, combining MEG/EEG with fMRI, through simultaneous EEG-fMRI acquisition or representational similarity analysis-based fusion approaches, could address the spatial resolution limitations inherent in MEG/EEG methods. Another promising direction is the development of neural modulation techniques, such as transcranial magnetic stimulation (TMS) or transcranial electrical stimulation (tES), to selectively manipulate periodic or aperiodic activity. These methods could provide causal evidence for the role of aperiodic components in cognitive processes.

An important methodological consideration for future EEG/MEG studies is the separation of aperiodic and oscillatory components. This distinction is critical for accurately interpreting neural dynamics, as failing to account for aperiodic activity can confound conclusions about oscillatory processes. Revisiting earlier findings with IRASA or similar approaches (e.g., fitting oscillations & one over f; FOOOF[10]) could yield valuable insights. For example, a recent study found that a considerable proportion of theta activity observed in working memory tasks may actually reflect shifts in aperiodic activity[62]. Similarly, another study reported that most functional connectivity patterns in resting-state EEG studies likely reflect aperiodic networks rather than oscillation-based networks[85]. These findings highlight the importance of adopting this methodological refinement to advance the understanding of neural dynamics and functional connectivity.

Investigating domain-general aperiodic and oscillatory responses in clinical populations, such as individuals with neuropsychiatric conditions, could also provide valuable insights into how these systems are disrupted in disease states. Similarly, lifespan studies examining these responses in children, adolescents, and older adults could reveal developmental

trajectories and age-related changes, contributing to a more comprehensive understanding of the role of aperiodic responses in cognitive control across the lifespan. Beyond theoretical contributions, our findings have significant potential for real-world applications. In cognitive assessment, aperiodic broadband power could serve as a biomarker for monitoring cognitive load or identifying individuals at risk for cognitive decline. In clinical diagnostics, these markers could complement imaging-based approaches to refine diagnosis and treatment monitoring in disorders affecting cognitive control. Finally, in the field of brain-computer interfaces, separating aperiodic and oscillatory signals might improve the precision of neural decoding for adaptive systems.

In conclusion, this study suggests that both aperiodic and oscillatory components of human electrophysiological signals may reflect domain-general responses to demand and change in task content. Notably, aperiodic broadband power showed the strongest domain-general properties with robust coding of both demand and content and cross-task generalisability of these signals. The aperiodic and oscillatory systems had distinct source distributions, with the aperiodic broadband power being widely distributed across the brain and partially overlapping with the domain-general MD network, while oscillatory power was mainly modulated in frontal or occipital areas. Theta and beta power shared similar cortical patterns, whereas alpha power exhibited reversed patterns relative to the other two. These findings provide insights into the neural responses for human domain-general cognition, emphasizing the potential role of frontoparietal E/I balance in supporting a range of cognitive processes. This contributes to bridging the gap between findings in human fMRI, non-human primate, and human electrophysiological studies.

## Methods
### Participants
Forty-seven healthy, right-handed participants were recruited from the local community and the online participant database (SONA) at the University of Cambridge. Four participants were excluded from formal analysis due to having behavioural accuracy more than 3 standard deviations below the mean. Hence 43 participants (age 18-39 years, 31 females and 12 males) entered the final analysis. All participants were native Mandarin speakers with normal or corrected to normal vision without history of neurological disorders. They gave written informed consent to participate in the study and were paid for their time. The experiment was approved by the Cambridge Psychology Research Ethics Committee. All ethical regulations relevant to human research participants were followed.

We recruited native Mandarin speakers in order to avoid word-length differences between the alphanumeric and colour WM tasks (i.e., remembering letters and colours), as word-length is known to affect working memory (WM) performance[86]. Mandarin speakers naturally encode English letters in English, but visually presented colours in Mandarin, and we further specified this strategy in our instructions to participants. We then chose English letters that are monosyllabic (e.g., "A" /eɪ/, "B" /biː/) and used colours with monosyllabic names in Mandarin, such as red (/hʊŋ/), yellow (/hwɑːŋ/) and purple (/dzɿ/). This design ensured that both types of WM stimuli—letters and colours—were encoded as monosyllabic terms by the participants, thereby minimizing phonetic word length effects and balancing cognitive demand between the alphanumeric and colour conditions[87,88].

### Task design and procedure
To examine potential domain-general activity related to cognitive demand across multiple tasks, we selected three cognitive tasks (a working memory task, WM; a switching task, SWIT; and a multi-source interference task; MSIT). We manipulated each to have different levels of cognitive demand (hard vs. easy) and different stimuli contents (alphanumeric vs. colour) (Fig. 1). As stimuli contents were blocked, we had six subtasks in total (3 tasks * 2 contents).

For the WM task, we used a modified Sternberg task[89]. After a fixation cross was shown for 1.5 +/−0.1 s, four items (letters for the alphanumeric

subtask and coloured circles for the colour subtask) were presented in a horizontal line at the centre of the screen for 0.3 s. In the hard condition the memory set consisted of four unique items, while the set in the easy condition consisted of two unique items flanked by a # (alphanumeric subtask) or black circles (colour subtask). Thus, the physical size and the visual content were the similar in both conditions. After a 2 s delay period, a probe appeared at the centre of the screen and the participant were required to press a button within 2 s to indicate whether the probe was a member of the memory set. We selected the letters from the list "B", "D", "G", "K", "P", "Q", and "R", and selected the colours from red, orange, yellow, green, blue, purple, and pink.

For the SWIT task, after a fixation cross was shown for 1.5 +/−0.1 s, one item (a two-digit number for the alphanumeric subtask and a coloured circle or triangle in the colour subtask) was presented at the centre of the screen within a square or a diamond for 3 s. For the alphanumeric subtask, if the stimulus was surrounded by a square, participants were instructed to indicate whether it was odd or even. If the stimulus was surrounded by a diamond, participants were required to indicate whether it could be divided by 3. For the colour subtask, participants were required to indicate whether it was blue or red if the stimulus was surrounded by a square, and to indicate whether it was a circle or a triangle if the stimulus was surrounded by a diamond. Switch trials (the rule for the present trial was different from the last trial) were considered as the hard condition and repeat trials (the rule for the present trial repeated the last trial) were considered as the easy condition. The stimuli were selected from 12, 13, 15, and 16 for the alphanumeric subtask and selected from blue circle, red circle, blue triangle, and red triangle for the colour subtask.

The MSIT task was used as an inhibitory control task. After a fixation cross was shown for 1.5 +/−0.1 s, a row of three items (digits for the alphanumeric subtask and coloured circles for the colour subtask) were presented at the centre of the screen for 3 s. Participants were required to identify the unique item among the three items and to press a button with one of three fingers as quickly as possible and within 3 s. The target item was "1", "2", or "3" for the alphanumeric subtask and was "red", "green", and "blue" for the colour subtask. Participants underwent a practice to match "1", "2", and "3" (or "red", "green", and "blue") to three buttons. In the hard (incongruent) condition, the target item was presented in the position incongruent with the original value (e.g., "1" or "red" presented at the third position), and always flanked by different interfering numbers or colours (e.g., "331" or "blue, blue, red"). In the easy (congruent) condition, the target item was presented in the position compatible with its original value and flanked by 0 or "X" in the alphanumeric subtask (e.g., "100" or "1XX") while flanked by black circles in the colour subtask (e.g., "red, black, black").

Participants performed two runs of each subtask, yielding a total of 12 runs for each participant. Each run included two blocks of the same subtask, with 36 trials per block for the WM and MSIT tasks. For the SWIT task, each block included 37 trials, but the first trial of each block was dropped, resulting in 36 trials per block for analysis. For all tasks, trials in each block were randomized and equally split between easy and hard trials. For the WM task, there were equal numbers of positive (probe on the list) and negative trials (probe not on the list) and these trials were randomly intermixed. Participants were allowed to take short breaks between blocks and between runs. The sequence of the three tasks (WM, SWIT, and MSIT) was counterbalanced across participants to control for order effects. Within each task, participants completed a run of each type of subtask once before repeating it. The order of subtask presentation was counterbalanced across participants.

### MEG/EEG data acquisition and preprocessing
In this study, we used combined MEG and EEG data to take advantage of the complementary strengths of these two modalities. MEG is selectively sensitive to tangential sources and is primarily sensitive to superficial cortical sources, while EEG detects both radial and tangential sources and is more sensitive to deep sources[90]. This complementarity can enhance decoding performance at the sensor level by providing richer information and can

reduce point-spread function, improve spatial resolution and localization accuracy at the source level[90–92].

MEG data were acquired in a magnetically shielded room with a Neuromag Vectorview system (Elekta AB, Stockholm, Sweden) with 306 channels (204 planar gradiometers and 102 magnetometers). EEG data were acquired concurrently using a 70-channel MEG-compatible EEG cap (EasyCap GmbH, Herrsching, Germany) with an extended 10–10 electrode layout. EEG reference and ground electrodes were attached to the left side of the nose and the left cheek, respectively. We recorded electrooculogram (EOG) by placing electrodes above and below the right eye (vertical EOG) and at the outer canthi (horizontal EOG). We also recorded electrocardiogram (ECG) by placing electrodes to the bottom left of ribcage and the right clavicle. For the co-registration with MRI structural data, we used a 3D digitizer (3SpaceIsotrakIISystem, Polhemus, Colchester, Vermont, USA) to record the positions of 5 head position indicator (HPI) coils, EEG electrodes, and 80–200 additional points covering the whole EEG cap, relative to three anatomic fiducial points (nasion and left and right preauricular points). Data were acquired with a sampling rate of 1000 Hz with a band-pass filter of 0.03–330 Hz.

We used MNE-Python[93] for all the processing steps. First, we applied the Signal-Space Separation (SSS) to the raw MEG data to reduce environmental artefacts[94]. This step also involved head movement compensation and bad channel repair for MEG data. Then, we visually inspected the raw data for each participant and marked bad EEG channels, which were excluded from further analysis. To reduce the eye movement and cardiac artefacts, we performed an independent component analysis (ICA) using the FastICA algorithm[95] for each participant after concatenating data from all the runs. Before running the ICA, the data was downsampled to 200 Hz and filtered between 1 and 40 Hz to improve ICA performance[96]. The first $n$ components explaining the cumulative variance of 95% of the data were entered into the ICA. We applied the computed ICA filters to the raw data (not down-sampled or filtered) and removed eye movement and heart-beat artefacts from both MEG and EEG. We identified EOG- and ECG-related components using correlations between components and the EOG/ECG channel, with components above the threshold (z-scores > 3) masked as EOG- or ECG-related and removed. After ICA, we filtered the data to 1–40 Hz, re-referenced the EEG data to the channel average, and divided data into epochs from −1 to 2 s from stimulus onset. We rejected epochs with excessive noise if their peak-to-peak amplitudes were higher than the following thresholds: 350 µV for EEG, 5000 fT for magnetometers, and 4000 fT/cm for gradiometers. Incorrect or overtime trials were also rejected. Using this procedure a total of 67 out of 868 trials (7.72%) on average were excluded from further analysis.

### Source reconstruction

For each participant, a T1-weighted MRI structural scan was acquired using a Siemens 3 T Prisma scanner. We used the automated segmentation algorithms in FreeSurfer (http://surfer.nmr.mgh.harvard.edu/) to obtain the reconstructed scalp surface[97]. Then, we used MNE-Python to compute forward and inverse models. We used a three-layer boundary element model (BEM) to compute the forward model of each participant's scalp, outer skull surface, and inner skull surface using default conductivity parameters (0.3, 0.006, and 0.3, respectively). Both MEG and EEG data were included in the forward model using the *mne.make_forward_solution* function, enabling the integration of the complementary sensitivity profiles of the two modalities. The MEG/EEG sensor configurations and MRI structural images were co-registered by matching the scalp digitization points to the scalp surface. For source analyses on task demand, we calculated noise covariance matrices for each participant and each task based on the pre-stimulus baseline period (−400 to −100 ms from the stimulus onset). For source analyses on task content, we calculated noise covariance matrices based on a 10-second period from the empty room data recorded on the testing day. This is because the task contents were blocked meaning the pre-stimulus time periods was not an ideal baseline in this case. Then, we selected the best estimator of the noise covariance matrices among a list of

methods ("shrunk", "empirical", "factor_analysis", and "diagonal_fixed") based on the log-likelihood and cross-validation separately for each participant and analysis (demand/content), using the automatic method embedded in *mne.compute_covariance* function[98]. The inverse operator was obtained for each participant and each task using dynamic statistical parametric mapping[99] (dSPM) with a loose orientation constraint value of 0.2 and without depth weighting. We used the default SNR = 3.0 to regularize the inverse operator. In order to perform group analyses, we morphed individual source space data to the standard average brain (fsaverage), yielding time courses of activity for 10,242 vertices in each hemisphere for each participant. We parcellated the standard fsaverage brain to 360 regions (180 regions per hemisphere without the medial wall) based on the Human Connectome Project multimodal parcellation (HCP-MMP1.0)[53] by averaging values of each vertex within each parcel using the *'mean_flip'* mode. The *'mean_flip'* mode finds the dominant direction of source space normal vector orientations for each parcel and applies a sign-flip to vertices whose orientation is more than 90° different from the dominant direction, and then averages across vertices within each parcel.

### Multivariate pattern analysis (MVPA)

For MVPA in sensor space, we used the data from all the MEG and EEG sensors (306 MEG sensors and 64 EEG channels). We used MNE-Python[93] and the Scikit-learn package[100] in Python for the analysis. To reduce computation time, we downsampled the data to 200 Hz. After removing the evoked potentials (i.e., the average signal across trials) from each trial for each condition[32], we used IRASA[50] to separate the oscillatory and the aperiodic components from the mixed power spectrum using the time window of 0.3–1.5 s relative to stimulus onset for each subtask. IRASA achieves this by resampling the original signal at slightly shifted rates (e.g., 1.1× to 1.9×), averaging the resulting power spectra to minimize periodic signals, and isolating the broadband aperiodic component. For our analysis, we used a recommended set of resampling factors ($h$) from 1.1 to 1.9 with the linear space of 0.05[50]. With the 200 Hz sampling rate and the maximum resampling factors we chose, frequencies up to 50 Hz were able to be estimated. Thus, we estimated the frequencies from 1 to 30 Hz with 1 Hz steps covering all the frequency bands of interest. After obtaining the aperiodic spectrum from IRASA, we subtracted it from the mixed spectrum to obtain the oscillatory spectrum. To obtain the oscillatory power for further analyses, we averaged the power for each frequency band using the ranges theta: 3–7 Hz; alpha: 8–12 Hz; and beta: 15-30 Hz. For aperiodic broadband power, we averaged the aperiodic spectrum power across 3–30 Hz. To obtain the slope and intercept of the aperiodic components, we used the least squares estimation to fit a linear function to the estimated aperiodic power spectrum in log-log coordinates, extracting the slope and the intercept for later analyses.

To increase the signal-to-noise ratio of MVPA, we averaged every 4 trials into "pseudo-trials" for each condition (e.g., hard or easy condition in the colour WM subtask) using a random selection of the available trials[101]. The choice of averaging every 4 trials was informed by previous research indicating that modest trial averaging (roughly 5-10%) provides an effective trade-off between improving signal-to-noise ratio and maintaining a sufficient number of trials for accurate classification[51,101]. When the number of trials was not divisible by 4, we excluded the remainder trials. If the number of resulting pseudo-trials was not the same for the two conditions, we equated them by removing pseudo-trials from the condition with more. Then, the data for each channel were standardized by subtracting the mean and dividing by the standard deviation across epochs. We then used a 5-fold cross-validation procedure for MVPA classification with a linear support vector machine (SVM) for each frequency band or aperiodic component, where the pseudo-trials were randomly divided into 5 parts, of which 4 were used for training and 1 for testing. The cross-validation procedure was repeated 5 times such that all pseudo-trials were used equally for training and testing. After splitting the data into folds, we performed PCA across sensors to retain components accounting for 99% of the variance and used those principal components for classification. Specifically, for each iteration,

we performed PCA on the training fold (i.e., 4 folds altogether) and then transformed the testing fold using the same PCA components. The decoding results (the area under the receiver-operating curve; AUC) across those 5 iterations were averaged together. To ensure the results were not dependent on a specific set of trial averages, we repeated the entire procedure 25 times per participant and averaged the AUC over these repetitions to give the final AUC for statistical testing. The MVPA procedure was performed on each subtask based on either oscillatory components or aperiodic components separately. We classified hard vs. easy trials for each subtask for demand decoding and classified alphanumeric vs. colour trials for each task for content decoding.

To estimate the source patterns underpinning each classification, we used the weight projection method[54]. We first repeated the same decoding analysis, now based on the 360 ROIs in source space, using the same procedure described above. Classification weights for each ROI indicated how much the information measured in that region was used by a classifier to separate the classes (e.g., hard vs. easy). To interpret these weights in a meaningful way, we transformed them to the activation patterns by multiplying the weights by the covariance of the data[54]. The transformed weights are equivalent to the univariate response in each contrast (e.g., hard vs. easy) and are informative to identify brain regions that contributed to the most to the classification. Positive patterns indicate increased activity in the hard condition (or the colour condition for content decoding) compared to the easy condition (or the alphanumeric condition for content decoding), and vice versa[54]. For averaged results across subtasks, we z-scored the data across all ROIs and then averaged the z-scored data across participants and subtasks.

### Cross-task generalisation
For the cross-task generalisation analysis in sensor space, we trained MVPA classifiers based on one subtask using all MEG/EEG sensors and then tested them in all other subtasks, yielding a training subtask * testing subtask matrix. We used the same decoding settings (e.g., pseudo-trials) as described above, except that here we used all the data from each subtask without splitting them into 5 folds. This procedure was done for each aperiodic and oscillatory signal separately. The off-diagonal results showed the average of the generalisation performance between two specific train-test schemes within a pair of subtasks (e.g., training on alphanumeric WM task, testing on colour SWIT task, and training on colour SWIT task, testing on alphanumeric WM task), with significantly above-chance AUC meaning the signal was generalisable across the two tasks.

### Representation geometry in the state space
Representation geometry analysis was done in sensor space. To visualize how the signals simultaneously represented task demand and content information, we first averaged trials within each condition (12 conditions in total: 3 tasks * 2 demands * 2 contents) for each sensor and each signal, then did PCA to extract the first two PCs for each condition and signal. To average results across tasks, we aligned each participant's PCs of different tasks using Procrustes transformation (without a scaling component). This step ensured that differences in rotation and translation did not distort comparisons across tasks. After averaging the aligned PCs across tasks, another Procrustes transformation without a scaling component was again used to align the PCs of different participants for visualizing the averaged results across participants.

### Statistics and reproducibility
All statistical analyses were performed in Python. We used repeated measures ANOVAs for the behavioural analysis, used Pearson's correlation for the correlation analysis, and used one-sample $t$ tests for all other tests. The Benjamini-Hochberg (BH) procedure was used to control the false discovery rate (FDR) for the $t$ tests for each analysis in the results, and we reported the significant results with corrected $p < 0.05$ for these tests. To test the significance of the mean correlation coefficient across participants, we conducted permutation tests with 1000 iterations. For each permutation, the labels of one variable were randomly shuffled within each participant's data. Pearson's correlation coefficients were computed for both the original and permuted data sets. We averaged correlation coefficients over participants per permutation, yielding a null distribution of permutated mean correlation coefficients across participants after 1000 permutations. The significance of the observed mean correlation was determined by comparing it to the null distribution generated from the permutations. One-tailed tests were used for the decoding analysis classifying task demand or content, since below-chance decoding is uninterpretable. Otherwise, two-tailed tests were used.

### Reporting summary
Further information on research design is available in the Nature Portfolio Reporting Summary linked to this article.

### Data availability
Source data used in this paper are available on the Open Science Framework (OSF) platform (https://osf.io/h2q6n/). Raw MEG/EEG data for all participants, along with other data, are available upon reasonable request from the MRC Cognition and Brain Sciences Unit data repository after publication at https://www.mrc-cbu.cam.ac.uk/publications/opendata/.

### Code availability
The code used for this study is available for download from the OSF platform at https://osf.io/h2q6n/.

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

## Acknowledgements
This project was supported by UKRI MRC intramural funding (SUAG/093/G116768) to A.W., and a Gates Cambridge Scholarship awarded to R.L. (OPP1144). The authors thank Olaf Hauk and Máté Aller for their suggestions on MEG/EEG data processing. For the purpose of open access, the author has applied a Creative Commons Attribution (CC BY) licence to any Author Accepted Manuscript version arising from this submission.

## Author contributions
R.L., and A.W. conceived the project; R.L., J.D., and A.W. designed the experiment; R.L., N.D., and A.W. implemented the experiment; R.L. conducted the experiment; R.L., N.D., and A.W. analysed data; R.L. wrote the first draft of the paper; R.L., N.D., J.D., and A.W. contributed to the final draft of the paper; A.W. provided overall supervision; and A.W. and R.L contributed funding.

## Competing interests
The authors declare no competing interests.
