## [Transparent Peer Review file · Communications Biology]

Aperiodic and oscillatory systems underpinning human domain-general cognition

Corresponding Author: Mr Runhao Lu

This manuscript has been previously reviewed at another journal. This document only contains information relating to versions considered at Communications Biology.

Version 0:

Reviewer comments:

Reviewer #1

(Remarks to the Author)

The authors used combined MEG and EEG recordings and multivariate classification strategies in healthy adults performing three cognitive tasks with two different levels of difficulty. The goal was to determine if decoding would reveal domain-general brain activity that coded 'cognitive load' independent of task (domain general) or code task content (task specific activity), and if this would differ for oscillatory activity (theta, alpha, beta) versus non-periodic activity (power and 1/f slope separated from oscillatory using the IRASA algorithm); the latter assumed to represent shifts in the balance between excitatory / inhibitory activity as opposed to task specific activation. They found that both types of neural activity coded for difficulty but with slightly different source distributions. Although their assumption that aperiodic (broadband) power is a 'proxy' of E/I balance based on ECoG recordings requires further validation for MEG/EEG recordings, this is an interesting approach to the study of role of cortical oscillations in cognitive processing which has to date focused on frequency specific encoding of mental operations.

The behavioural analysis of task performance and its relationship to the classification analysis is well described, however the manuscript would benefit from some clarifications of the EEG and MEG methods.

1. It is never made particularly clear why the authors decided to use a combination of EEG and MEG, or when and how these two modalities were combined and in particular how this should be interpreted in terms of the cortical distribution of activity related to better or worse decoding. At the sensor level one might assume that the more information the classifier has the better the decoding. However, this would also confound the types of brain activity measured by these two modalities.

They have very different sensitivities to different portions of the cortex, and cortical versus subcortical activity. Did the authors try decoding with just EEG or just MEG to see if they get a similar result? If so this would suggest the methods are providing redundant information for the decoder. If not, this would be informative in terms of whether the domain general activity is e.g., dominated by radial generators in the EEG which could mask much of the MEG (sulcal) activity in primary sensory areas etc. Similarly, the two methods may differ in terms of sensitivity to E/I type mechanisms. Have any previous studies on separating periodic and non-periodic activity combined MEG and EEG in this way?

2. Why is the decoding done at both the sensor and source level? Perhaps I am misreading the description of the methods but on page 23 it reads as if both EEG and MEG data combined to reconstruct source activity at the level of cortical parcels? How and why was this done? MEG and EEG have very different spatial resolutions and relative accuracy in terms of co-registering activity with brain anatomy. EEG is also much more susceptible to artifacts, particularly muscle artifacts. How might this bias the estimation of broadband power?

In general, some rationale of why using both EEG and MEG modalities and why they are not tested separately is needed.

Minor comments:

- What is the rationale for making 'pseudo-trials' by averaging 4 single trials? The increase in SNR would be rather small. Was this based on some previous work?

- The explanation of the choice of native Mandarin speakers and task design is a bit confusing. It is argued that this was to equate 'phonetic' word length effects between English and Mandarin. But the tasks appear to be using English letters and numbers not Mandarin symbols. Is the assumption that the Mandarin speakers are encoding the letters in their native language? Please clarify?

Reviewer #2

(Remarks to the Author)

In the manuscript, the authors evaluate the roles aperiodic and oscillatory components of magnetoencephalography (MEG) and electroencephalography (EEG) time series data play in domain-general cognition. Analyzing the MEG/EEG data recorded during three cognitive tasks with sophisticated methods, they reported that task demand and content could be decoded from parameters of both aperiodic and oscillatory activities successfully, but the spatial patterns of their sources differed. Using representational geometry analysis, the authors further revealed that task demand and content representation had clear boundaries between their conditions. The authors discuss the electrophysiological mechanisms underlying the human domain-general cognition with the recent findings that aperiodic activity may reflect the balance of excitatory and inhibitory neural activity (so-called E/I balance).

The topic of aperiodic components and E/I balance in human electrophysiological activity is timely, and the present study is very interesting. I have a few minor comments on the manuscript, and specific points to be addressed are as follows.

1. "Then, we applied the IRASA [53] on the time windows of 0.3-1.5 second from stimulus onset for each task to separate the aperiodic and oscillatory components from the mixed signal (Figure 2A)." (P. 6, lines 14–16):

I wonder why the authors excluded the time windows of 0-0.3 second from stimulus onset when applying the IRASA to obtain the aperiodic components in the frequency domain. There seems to be no problem to include the period if evoked responses were removed by subtraction. Please explain this point.

2. Figure 2:

It would be better to reverse the order in which figures 2C and 2D are placed or the order of the corresponding sentences in the main text.

3. Figure 3:

Similar to the above comment 2, I think it would be better to change the order in which figures 3B, 3C and 3D are placed or the order of the corresponding sentences in the main text.

Reviewer #3

(Remarks to the Author)

Introduction

"However, it is not known whether corresponding domain-general electrophysiological responses, such as those measured by magnetoencephalography (MEG) or electroencephalography (EEG), support different types of cognitive activity." What do you mean by this? What cognitive activity here you're referring to? What is the differentiation between these and previously mentioned "a variety of cognitive tasks"?

I highly suggest you rephrase the logic in the first paragraph, which seems to be repetitive and double-layered.

The role of the paragraph starting from line 61 seems not clear. Could you try to make it better in transitioning between paragraphs?

Discussion

Regarding the discussion section, I have following aspects of suggestions:

1. Expand on how these findings relate to major cognitive control theories, such as the adaptive coding model or the multiple demand theory. Discuss how the aperiodic and oscillatory components might fit into or challenge existing frameworks.
2. Highlight the importance of separating aperiodic and oscillatory components in future EEG/MEG studies. Discuss how this approach might change interpretations of previous research that didn't make this distinction.
3. Include a paragraph discussing potential limitations of the study, such as sample size, task specificity, or technical constraints of the MEG/EEG methods used.
4. Propose specific follow-up studies or experiments that could further validate or extend these findings. For example, suggest investigating these patterns in clinical populations or during different cognitive states.
5. Discuss potential real-world applications of these findings, such as in cognitive assessment, brain-computer interfaces, or clinical diagnostics.
6. Elaborate on how these MEG/EEG findings complement or contrast with results from other neuroimaging techniques like fMRI or PET.
7. Address whether there were notable individual differences in these domain-general responses and what factors might contribute to such variations.
8. Consider discussing how these domain-general aperiodic and oscillatory responses might develop over the lifespan or differ in various age groups.

The results and method are relatively good. However, if the terms and calculations could be further clarified to be understandable to outsiders, the article would be further bettered.

Version 1:

Reviewer comments:

Reviewer #1

(Remarks to the Author)

The authors have addressed my concerns in the text of the article and have performed some additional analysis of the data to confirm whether EEG or MEG alone would also result in successful decoding. This will help clarify this point for the readers.

Reviewer #2

(Remarks to the Author)

Reviewer #3

(Remarks to the Author)

I endorse the publication of this article!

Dear Editors:

We would like to sincerely thank you and the reviewers for their insightful comments and suggestions, which we believe have clearly improved the quality of our manuscript. In response to the reviewers' concerns, we have edited the manuscript as detailed below (our responses highlighted in blue).

Sincerely yours,

Runhao Lu, Dr. Nadene Dermody, Prof. John Duncan, & Prof. Alex Woolgar

Reviewer #1 (Remarks to the Author):

The authors used combined MEG and EEG recordings and multivariate classification strategies in healthy adults performing three cognitive tasks with two different levels of difficulty. The goal was to determine if decoding would reveal domain-general brain activity that coded 'cognitive load' independent of task (domain general) or code task content (task specific activity), and if this would differ for oscillatory activity (theta, alpha, beta) versus non-periodic activity (power and 1/f slope separated from oscillatory using the IRASA algorithm); the latter assumed to represent shifts in the balance between excitatory / inhibitory activity as opposed to task specific activation. They found that both types of neural activity coded for difficulty but with slightly different source distributions. Although their assumption that aperiodic (broadband) power is a 'proxy' of E/I balance based on ECoG recordings requires further validation for MEG/EEG recordings, this is an interesting approach to the study of role of cortical oscillations in cognitive processing which has to date focused on frequency specific encoding of mental operations. The behavioural analysis of task performance and its relationship to the classification analysis is well described, however the manuscript would benefit from some clarifications of the EEG and MEG methods.

1. It is never made particularly clear why the authors decided to use a combination of EEG and MEG, or when and how these two modalities were combined and in particular how this should be interpreted in terms of the cortical distribution of activity related to better or worse decoding.

At the sensor level one might assume that the more information the classifier has the better the decoding. However, this would also confound the types of brain activity measured by these two modalities. They have very different sensitivities to different portions of the cortex, and cortical versus subcortical activity. Did the authors try decoding with just EEG or just MEG to see if they get a similar result? If so this would suggest the methods are providing redundant information for the decoder. If not, this would be informative in terms of whether the domain general activity is e.g., dominated by radial generators in the EEG which could mask much of the MEG (sulcal) activity in primary sensory areas etc. Similarly, the two methods may differ in terms of sensitivity to E/I type mechanisms. Have any previous studies on separating periodic and non-periodic activity combined MEG and EEG in this way?

Responses:

We thank the reviewer's detailed and insightful comments. Below, we clarify the rationale for combining EEG and MEG, the outcomes of single-modality decoding, and the interpretation of our results with respect to E/I mechanisms.

(1) Why combine MEG and EEG?

The decision to use both MEG and EEG stems from the complementary nature of these modalities, both at the sensor and source levels:

At the sensor level, our aim is to examine whether oscillatory and aperiodic brain activity contains task-relevant demand and content information. Thus, it is helpful to have richer information by including all available data.

At the source level, source reconstruction benefits significantly from combining modalities due to their different sensitivities (Samuelsson, Khan, Sundaram, Peled, & Hamalainen, 2019) (also see Response Figure 1 and 2). As shown in previous studies (Henson, Mouchlianitis, & Friston, 2009; Molins, Stufflebeam, Brown, & Hamalainen, 2008) and corroborated in the MNE-Python demo script (https://mne.tools/stable/auto_examples/inverse/resolution_metrics_eegmeg.html), combining EEG and MEG reduces the point-spread function, increases spatial resolution and localization accuracy, especially for deeper sources. This enhancement arises from the complementary sensitivity profiles of EEG and MEG (e.g., MEG is more sensitive to tangential currents, while EEG captures both tangential and radial currents), which jointly mitigate biases introduced by channel-specific properties.

Importantly, at the source level, results are interpreted based on the unified source space generated

from both modalities (also see our response to the next question), rather than individual channel contributions. This integration emphasizes complementarity rather than confound.

(2) Decoding with EEG or MEG alone

To investigate whether the two modalities provide similar results, we performed decoding analyses using EEG-only and MEG-only channels (see Response Figure 3, sensor-level decoding of task demand). The results were broadly similar across the modalities; however, the combined MEG-EEG data showed slightly higher decoding performance than either MEG alone or EEG alone. These results suggest that while both modalities independently capture task demand information, their combination provides additional discriminative power. Moreover, MEG and EEG signals are not redundant at the source level, because their complementary spatial sensitivity could contribute to improved spatial resolution when combined both modalities.

(3) Sensitivity to E/I mechanisms

To our knowledge, no previous studies have specifically investigated the separation of periodic and non-periodic activity using combined MEG and EEG data in this way. However, the general rationale for combining these modalities remains robust. At the sensor level, while MEG and EEG may differ in sensitivity to E/I mechanisms, integrating both modalities ensures that we have richer neural information to enhance decoding performance, answering the question about if oscillatory/apperiodic activity represents demand/content information. At the source-level, both MEG and EEG signals were combined to generate a unified source space, where we performed decoding analysis. This approach uses complementary sensitivities of MEG and EEG while mitigating potential biases introduced by their systematic differences. Also, the multivariate decoding method used in our study is robust to handle potential systematic differences between modalities, especially since we standardized the data across trials for each channel type separately.

In summary, combining MEG and EEG is justified by their complementary sensitivity profiles, which can enhance decoding performance and improve spatial resolution for source reconstruction. We have added a paragraph into the Methods part to clarify our rationale (Lines 615-620).

Lines 615-620:

“In this study, we used combined MEG and EEG data to take advantage of the complementary strengths of these two modalities. MEG is selectively sensitive to tangential sources and is primarily

sensitive to superficial cortical sources, while EEG detects both radial and tangential sources and is more sensitive to deep sources [90]. This complementarity can enhance decoding performance at the sensor level by providing richer information and can reduce point-spread function, improve spatial resolution and localization accuracy at the source level [90-92].”

Response Figure 1: Sensitivity map for gradiometer, magnetometer, and EEG channels. MEG channels (gradiometers and magnetometers) show high sensitivity to superficial sources, while EEG channels have broader sensitivity, including deeper sources. Adapted from Samuelsson et al. (2019).

Response Figure 2: Sensitivity map for gradiometer, magnetometer, and EEG channels calculated from one participant in the present study, showing similar patterns to those reported in Samuelsson et al., (2019). See MNE-Python documentation for details about how these sensitivity maps were calculated.

Response Figure 3: Decoding results on task demand (hard vs. easy) using aperiodic or oscillatory activity for each subtask using MEG or EEG sensors separately. Error bars represent standard errors.

*** $p < 0.001$ (FDR-corrected)

2. Why is the decoding done at both the sensor and source level? Perhaps I am misreading the description of the methods but on page 23 it reads as if both EEG and MEG data combined to reconstruct source activity at the level of cortical parcels? How and why was this done? MEG and EEG have very different spatial resolutions and relative accuracy in terms of co-registering activity with brain anatomy. EEG is also much more susceptible to artifacts, particularly muscle artifacts. How might this bias the estimation of broadband power? In general, some rationale of why using both EEG and MEG modalities and why they are not tested separately is needed.

Responses:

We thank the reviewer for the comments. Below, we address each concern in detail:

(1) Why Decode at Both Sensor and Source Levels?

As noted in our response to the previous question, sensor-level decoding uses all available data to explore whether task-relevant demand/content information is coded in oscillatory or aperiodic signals, while source-level decoding identifies the cortical regions most involved in encoding this information. This dual approach provides complementary insights into both the "what" and "where" of neural processes.

(2) Combining MEG and EEG Signals for Source Reconstruction

Yes, we indeed combined MEG and EEG signals to reconstruct the source space, and this is a standard approach in concurrent MEG-EEG studies. Toolboxes such as MNE-Python readily support this (e.g., function `mne.make_forward_solution()` with `MEG=True` and `EEG=True`), enabling integration of both modalities in the forward model. As you mentioned, MEG and EEG have different spatial sensitivity, and the combination therefore enhances spatial resolution by leveraging MEG's sensitivity to tangential currents and EEG's sensitivity to both tangential and radial currents (Molins et al., 2008).

(3) Artifact handling in EEG data

Although EEG data is more susceptible to artifacts, we followed standard preprocessing pipelines ensuring high signal-to-noise ratio. The butterfly plot of EEG signals (Response Figure 4) confirms the absence of significant residual artifacts, minimizing the risk of bias in broadband power estimation.

(4) Why not test EEG and MEG separately?

As noted in the previous response, our aim was to maximize sensitivity by combining these complementary sources of information. However, in response to your question, we performed decoding analyses separately for EEG and MEG data (Response Figure 4). The results were broadly similar, but the combined MEG-EEG data yielded slightly better performance, underscoring their complementary contributions. Also, the combined modality is particularly advantageous for source reconstruction, improving spatial resolution and localization accuracy.

We have added a paragraph in Methods to clarify the rationale for using combined MEG and EEG data (Lines 615-620) and how we use both MEG and EEG data for source reconstruction (Lines 657-659).

Lines 615-620:

“In this study, we used combined MEG and EEG data to take advantage of the complementary strengths of these two modalities. MEG is selectively sensitive to tangential sources and is primarily sensitive to superficial cortical sources, while EEG detects both radial and tangential sources and is more sensitive to deep sources [90]. This complementarity can enhance decoding performance at the sensor level by providing richer information and can reduce point-spread function, improve spatial resolution and localization accuracy at the source level [90-92].”

Lines 657-659:

“Both MEG and EEG data were included in the forward model using the *mne.make_forward_solution* function, enabling the integration of the complementary sensitivity profiles of the two modalities.”

Response Figure 4: Butterfly plots for EEG, gradiometer, and magnetometer channels from three exemplar participants, showing grand-averaged event-related responses across all trials.

Minor comments:

3. What is the rationale for making ‘pseudo-trials’ by averaging 4 single trials? The increase in SNR would be rather small. Was this based on some previous work?

Responses: We thank the reviewer for this comment. Yes, the choice of making ‘pseudo-trials’ by averaging 4 trials is indeed based on previous literature (Grootswagers, Wardle, & Carlson, 2017; Scrivener, Grootswagers, & Woolgar, 2023). Grootswagers et al. (2017) found that averaging 4 trials in their exemplar data yielded a substantial improvement in SNR and decoding performance, but averaging more trials does not increase decoding performance by the same factor. This result suggested

that averaging a modest number of trials (e.g. 4 trials) is a reasonable balance between enhancing SNR and retaining sufficient trial numbers. Similarly, Scrivener et al. (2023) recommended that modest trial averaging using roughly 5-10% of the total number of trials per condition improved decoding performance and associated t-statistics. In our study, with 72 trials per condition and approximately 66 trials on average after preprocessing, averaging four trials (roughly 6%) falls within this recommended range. We believe that this approach allowed us to improve SNR while preserving enough trials per condition for robust classification. We have clarified this in the revised manuscript (Lines 702-705)

Lines 702-705:

“The choice of averaging every 4 trials was informed by previous research indicating that modest trial averaging (roughly 5-10%) provides an effective trade-off between improving signal-to-noise ratio and maintaining a sufficient number of trials for accurate classification [51, 101].”

4. The explanation of the choice of native Mandarin speakers and task design is a bit confusing. It is argued that this was to equate ‘phonetic’ word length effects between English and Mandarin. But the tasks appear to be using English letters and numbers not Mandarin symbols. Is the assumption that the Mandarin speakers are encoding the letters in their native language? Please clarify?

Responses: Thank you for raising this point. To clarify, we used only English letters and visually presented colours (as coloured circles) as stimuli in our WM tasks (no numbers). Mandarin speakers naturally encode letters in English but encode colours in Mandarin, and we specified this in our instructions to ensure consistency across participants. In our study, we selected English letters that were all monosyllabic (e.g., A /ei/, B /bi:/), and used colours with monosyllabic names in Mandarin, such as yellow (/hwa:ŋ/) and purple (/dzi/). This design ensured that both types of stimuli in the working memory task—letters and colours—were encoded as monosyllabic terms in Mandarin, thereby minimizing phonetic word length effects and balancing cognitive load between the alphanumeric and color conditions. We have clarified that in the manuscript (Lines 556-564).

Lines 556-564: “We recruited native Mandarin speakers in order to avoid word-length differences between the alphanumeric and colour WM tasks (i.e., remembering letters and colours), as word-length

is known to affect working memory (WM) performance [86]. Mandarin speakers naturally encode English letters in English, but visually presented colours in Mandarin, and we further specified this strategy in our instructions to participants. We then chose English letters that are monosyllabic (e.g., “A” /eɪ/, “B” /bi:/) and used colours with monosyllabic names in Mandarin, such as red (/hʊŋ/), yellow (/hwa:ŋ/) and purple (/dzi/). This design ensured that both types of WM stimuli—letters and colours—were encoded as monosyllabic terms by the participants, thereby minimizing phonetic word length effects and balancing cognitive demand between the alphanumeric and colour conditions [87, 88]”

Reviewer #2 (Remarks to the Author):

In the manuscript, the authors evaluate the roles aperiodic and oscillatory components of magnetoencephalography (MEG) and electroencephalography (EEG) time series data play in domain-general cognition. Analyzing the MEG/EEG data recorded during three cognitive tasks with sophisticated methods, they reported that task demand and content could be decoded from parameters of both aperiodic and oscillatory activities successfully, but the spatial patterns of their sources differed. Using representational geometry analysis, the authors further revealed that task demand and content representation had clear boundaries between their conditions. The authors discuss the electrophysiological mechanisms underlying the human domain-general cognition with the recent findings that aperiodic activity may reflect the balance of excitatory and inhibitory neural activity (so-called E/I balance). The topic of aperiodic components and E/I balance in human electrophysiological activity is timely, and the present study is very interesting. I have a few minor comments on the manuscript, and specific points to be addressed are as follows.

1. “Then, we applied the IRASA [53] on the time windows of 0.3-1.5 second from stimulus onset for each task to separate the aperiodic and oscillatory components from the mixed signal (Figure 2A).” (P. 6, lines 14–16): I wonder why the authors excluded the time windows of 0-0.3 second from stimulus onset when applying the IRASA to obtain the aperiodic components in the frequency domain. There seems to be no problem to include the period if evoked responses were removed by subtraction. Please explain this point.

Responses: Thank you for raising this question. We chose to exclude the initial 0-0.3 seconds after stimulus onset when applying the IRASA for several reasons: (1) In the working memory (WM) tasks,

we aimed to examine load effects during the maintenance period, which begins 0.3 seconds after stimulus presentation. The first 0.3 seconds post-stimulus onset correspond to the stimulus presentation period with WM items still visible on the screen, making it less suitable for analysing load-related activity. (2) Previous literature suggests that demand effects in the Switching (SWIT) and Multi-Source Interference (MSIT) tasks tend to stabilize after approximately 300 ms, as the period before this may primarily be associated with perceptual processing (Bush & Shin, 2006; Proskovec, Wiesman, & Wilson, 2019). Excluding this initial period allowed us to focus on the neural dynamics more directly related to cognitive demand. We have explained this point in the manuscript (Lines 147-150).

Lines 147-150: “We excluded the initial 0-0.3 seconds to focus on the neural dynamics more directly related to cognitive demand, as the early window may primarily reflect perceptual activity. For the WM tasks, excluding this period allowed us to specifically analyse the maintenance period rather than the stimulus presentation period.”

2. Figure 2: It would be better to reverse the order in which figures 2C and 2D are placed or the order of the corresponding sentences in the main text.

Responses: Thank you for this suggestion. We have reversed the order of the corresponding paragraphs in the main text to fit the order of Figure 2C and 2D.

3. Figure 3: Similar to the above comment 2, I think it would be better to change the order in which figures 3B, 3C and 3D are placed or the order of the corresponding sentences in the main text.

Responses: Thank you for this suggestion. We have also reversed the order of the corresponding paragraphs in the main text to fit the order of Figure 3.

Reviewer #3 (Remarks to the Author):

Introduction:

1. “However, it is not known whether corresponding domain-general electrophysiological responses, such as those measured by magnetoencephalography (MEG) or electroencephalography (EEG), support different types of cognitive activity.” What do you mean

by this? What cognitive activity here you're referring to? What is the differentiation between these and previously mentioned "a variety of cognitive tasks"? I highly suggest you rephrase the logic in the first paragraph, which seems to be repetitive and double-layered.

Responses: We appreciate the reviewer's comment and the opportunity to clarify our intended meaning. In the original text, "different types of cognitive activity" referred to the same concept as "a variety of cognitive tasks," such as working memory, task switching, and inhibitory control. Our intention was to emphasize that while previous research has demonstrated the domain-general engagement of the frontoparietal regions in fMRI studies across such tasks, it remains unclear whether similar domain-general responses are observed in electrophysiological signals (MEG/EEG). To address this, we have rephrased the first paragraph to eliminate redundancy and clarify this distinction (Lines 24-33).

Lines 24-33: "Human cognition is accomplished by the joint contributions of both domain-general and highly specialized neural systems [1-3]. A specific set of frontoparietal regions, collectively referred to as the multiple-demand (MD) network, is known to support domain-general cognition by being consistently co-activated during a variety of cognitive tasks, such as working memory, task switching, inhibitory control, and many more [4-9]. While the MD network's engagement across diverse tasks is well-established in fMRI studies, it remains unclear whether there are corresponding domain-general electrophysiological responses—measured using magnetoencephalography (MEG) or electroencephalography (EEG)—that show similar generalization across these tasks. Investigating this question is important, as electrophysiological signals directly reflect neuronal activity, providing an essential bridge between electrophysiological findings and fMRI results."

2. The role of the paragraph starting from line 61 ('In addition to...') seems not clear. Could you try to make it better in transitioning between paragraphs?

Responses: We thank the reviewer's feedback regarding the role of this paragraph and its transition. This paragraph was intended to emphasize that the domain-general system (e.g., the MD network in fMRI literature), in addition to responding to various task demand, encodes a variety of task-relevant information such as task contents and rules. This concept serves as the foundation for the subsequent research question, which asks whether aperiodic/oscillatory signals exhibit a similar capacity for coding *both* task demand and content in a domain-general manner. We have revised the paragraph to

make this connection clearer and to improve the flow between sections (Lines 60-66).

Lines 60-66:

“In addition to responding more strongly to difficult compared to easy cognitive tasks, the domain-general MD network is also known to encode a broad range of task-relevant information such as stimuli, rules and responses [1, 46-49]. This suggests that the MD network represents multiple aspects of a task, rather than simply reflecting cognitive effort [46]. Building on this, it is important to ask whether the electrophysiological counterparts of the domain-general system also exhibit this versatility of encoding. Specifically, beyond coding task demand, do these signals also encode other information, such as the task content, in a manner consistent with domain-general functionality?”

3. Discussion: Regarding the discussion section, I have following aspects of suggestions:

(1) Expand on how these findings relate to major cognitive control theories, such as the adaptive coding model or the multiple demand theory. Discuss how the aperiodic and oscillatory components might fit into or challenge existing frameworks.

Responses: We appreciate the reviewer’s suggestion to expand on how our findings relate to major cognitive control theories. In response, we have added a section in the Discussion explicitly connecting our findings to the multiple demand (MD) theory and the adaptive coding hypothesis. (Lines 368-379).

Lines 368-379: “Our findings provide novel support for the MD theory [1, 5] by demonstrating that a widely distributed domain-general system can be detected in electrophysiological responses. This system, mainly reflected in aperiodic broadband power, exhibits robust generalization across tasks and coding of both task demand and task content. These results extend previous work in fMRI suggesting that human cognition is implemented by both domain-general and domain-specific systems [2, 3], which collaboratively enable the flexible cognitive control required for diverse tasks. Additionally, our findings align with the adaptive coding hypothesis, which posits that domain-general systems flexibly encode various types of task-relevant information [46, 47]. In particular, we observed that domain-general aperiodic activity and oscillatory power can represent both task demand and task content, albeit along different coding dimensions. This distinction highlights the adaptive capacity of domain-general systems to dynamically represent multiple aspects of a task, supporting the hypothesis that these

systems function as a flexible neural substrate for cognitive control.”

(2) Highlight the importance of separating aperiodic and oscillatory components in future EEG/MEG studies. Discuss how this approach might change interpretations of previous research that didn't make this distinction.

Responses: Thank you for this suggestion, we have added this point into the paragraph discussing future directions (Lines 513-522).

Lines 513-522: “An important methodological consideration for future EEG/MEG studies is the separation of aperiodic and oscillatory components. This distinction is critical for accurately interpreting neural dynamics, as failing to account for aperiodic activity can confound conclusions about oscillatory processes. Revisiting earlier findings with IRASA or similar approaches (e.g., fitting oscillations & one over f ; FOOOF [10]) could yield valuable insights. For example, a recent study found that a considerable proportion of theta activity observed in working memory tasks may actually reflect shifts in aperiodic activity [62]. Similarly, another study reported that most functional connectivity patterns in resting-state EEG studies likely reflect aperiodic networks rather than oscillation-based networks [85]. These findings highlight the importance of adopting this methodological refinement to advance the understanding of neural dynamics and functional connectivity.”

(3) Include a paragraph discussing potential limitations of the study, such as sample size, task specificity, or technical constraints of the MEG/EEG methods used.

Responses: We have added a paragraph to the Discussion section outlining key limitations (Lines 485-499).

Lines 485-499: “We acknowledge several limitations in this study. First, although the sample size was sufficient for group-level statistical analyses, we had limited ability to address questions regarding individual differences. Recruiting a larger and more diverse group of participants in future studies could provide a more nuanced understanding of variability across individuals. For instance, factors such as fluid intelligence may influence attentional allocation policies and thereby modulate the

observed domain-general responses [69, 83, 84]. Exploring such relationships in larger samples could help uncover how individual differences shape these neural patterns. Second, while the cognitive tasks we used captured a range of cognitive control processes, they do not capture the full range of cognitive domains. Incorporating more diverse and ecologically valid tasks could strengthen the applicability of these findings to broader cognitive contexts. Third, the limited spatial resolution of MEG/EEG constrained the precision of our source localization. This underscores the importance of integrating multimodal methods, such as fMRI, to enhance spatial specificity. Finally, although there is evidence linking aperiodic activity to E/I balance, finer mechanistic interpretations on these findings remain challenging. Addressing this will require dedicated computational modelling and systematic experiments at both intracranial and extracranial recording levels.”

(4) Propose specific follow-up studies or experiments that could further validate or extend these findings. For example, suggest investigating these patterns in clinical populations or during different cognitive states.

Responses: We have incorporated these points into a paragraph discussing future directions (Lines 500-512 and Lines 523-528).

Lines 500-512: “Building on our findings, several future directions could further validate and extend this work. A critical next step involves using multimodal approaches to deepen our understanding of the mechanisms underlying aperiodic and oscillatory activity. For instance, comparing MEG/EEG results with intracranial EEG data could provide crucial insights. The hypothesis linking aperiodic activity to E/I balance was largely developed with local field potential data, and confirming comparable patterns between intracranial EEG and MEG/EEG would strengthen the validity of non-invasive measures in assessing E/I balance and its role in cognition. Additionally, combining MEG/EEG with fMRI, through simultaneous EEG-fMRI acquisition or representational similarity analysis-based fusion approaches, could address the spatial resolution limitations inherent in MEG/EEG methods. Another promising direction is the development of neural modulation techniques, such as transcranial magnetic stimulation (TMS) or transcranial electrical stimulation (tES), to selectively manipulate periodic or aperiodic activity. These methods could provide causal evidence for the role of aperiodic components in cognitive processes.”

Lines 523-528: “Investigating domain-general aperiodic and oscillatory responses in clinical populations, such as individuals with neuropsychiatric conditions, could also provide valuable insights into how these systems are disrupted in disease states. Similarly, lifespan studies examining these responses in children, adolescents, and older adults could reveal developmental trajectories and age-related changes, contributing to a more comprehensive understanding of the role of aperiodic responses in cognitive control across the lifespan.”

(5) Discuss potential real-world applications of these findings, such as in cognitive assessment, brain-computer interfaces, or clinical diagnostics.

Responses: We have incorporated this point into a paragraph discussing future directions and potential real-world applications. (Lines 528-534).

Lines 528-534: “Beyond theoretical contributions, our findings have significant potential for real-world applications. In cognitive assessment, aperiodic broadband power could serve as a biomarker for monitoring cognitive load or identifying individuals at risk for cognitive decline. In clinical diagnostics, these markers could complement imaging-based approaches to refine diagnosis and treatment monitoring in disorders affecting cognitive control. Finally, in the field of brain-computer interfaces, separating aperiodic and oscillatory signals might improve the precision of neural decoding for adaptive systems.”

(6) Elaborate on how these MEG/EEG findings complement or contrast with results from other neuroimaging techniques like fMRI or PET.

Responses: Thank you for this suggestion. We have expanded the Discussion to address how our MEG/EEG findings complement with fMRI or PET results (Lines 413-420).

Lines 413-420: “These findings highlight how MEG/EEG measures of aperiodic activity may complement fMRI or PET by offering a window into neuronal dynamics, and potentially shifts in E/I balance, that may underlie domain-general cognition. While both modalities capture domain-general systems, they may reflect distinct yet interconnected processes: fMRI and PET measure metabolic

demands and regional activations, whereas MEG/EEG provides insights into electrophysiological activity including excitation and inhibition. Together, these perspectives enhance our understanding of how domain-general systems operate across multiple aspects of neural activity.”

(7) Address whether there were notable individual differences in these domain-general responses and what factors might contribute to such variations.

Responses: As mentioned in the limitation, the sample size limits our ability to address questions regarding individual differences. However, we discussed that factors such as fluid intelligence may influence attentional allocation policies and thereby modulate the observed domain-general responses [69, 83, 84]. We have incorporated this point into a paragraph discussing limitation. (Lines 485-491, see our reply to the point 3 above).

(8) Consider discussing how these domain-general aperiodic and oscillatory responses might develop over the lifespan or differ in various age groups.

Responses: We have incorporated this point into a paragraph discussing future directions. (Lines 525-528, see our reply to point 4 above).

4. The results and method are relatively good. However, if the terms and calculations could be further clarified to be understandable to outsiders, the article would be further bettered.

Responses: We thank the reviewer for this suggestion. We have reviewed the manuscript to make key terms and calculations explained clearly. We have added brief definitions for specialized terms (e.g., IRASA and MVPA) where they first appear, and also added more descriptions in the Methods (highlighted in blue). Also, we will make the scripts used for calculations publicly available online alongside publication of the paper, which will allow readers to follow or replicate our analysis.

References

- Bush, G., & Shin, L. M. (2006). The Multi-Source Interference Task: an fMRI task that reliably activates the cingulo-frontal-parietal cognitive/attention network. *Nat Protoc*, *1*(1), 308-313. doi:10.1038/nprot.2006.48
- Grootswagers, T., Wardle, S. G., & Carlson, T. A. (2017). Decoding Dynamic Brain Patterns from Evoked Responses: A Tutorial on Multivariate Pattern Analysis Applied to Time Series Neuroimaging Data. *J Cogn Neurosci*, *29*(4), 677-697. doi:10.1162/jocn_a_01068

- Henson, R. N., Mouchlianitis, E., & Friston, K. J. (2009). MEG and EEG data fusion: simultaneous localisation of face-evoked responses. *Neuroimage*, *47*(2), 581-589. doi:10.1016/j.neuroimage.2009.04.063
- Molins, A., Stufflebeam, S. M., Brown, E. N., & Hamalainen, M. S. (2008). Quantification of the benefit from integrating MEG and EEG data in minimum l2-norm estimation. *Neuroimage*, *42*(3), 1069-1077. doi:10.1016/j.neuroimage.2008.05.064
- Proskovec, A. L., Wiesman, A. I., & Wilson, T. W. (2019). The strength of alpha and gamma oscillations predicts behavioral switch costs. *Neuroimage*, *188*, 274-281. doi:10.1016/j.neuroimage.2018.12.016
- Samuelsson, J. G., Khan, S., Sundaram, P., Peled, N., & Hamalainen, M. S. (2019). Cortical Signal Suppression (CSS) for Detection of Subcortical Activity Using MEG and EEG. *Brain Topogr*, *32*(2), 215-228. doi:10.1007/s10548-018-00694-5
- Scrivener, C. L., Grootswagers, T., & Woolgar, A. (2023). Optimising analysis choices for multivariate decoding: Creating pseudotrials using trial averaging and resampling. *BioRxiv*. doi:10.1101/2023.10.04.560678